# An Inductive Bias for Tabular Deep Learning

**Ege Beyazit**[*]
Amazon
beyazit@amazon.com

**Jonathan Kozaczuk**[*]
Amazon
jonkozac@amazon.com

**Bo Li**
Amazon
booli@amazon.com

**Vanessa Wallace**
Amazon
vwall@amazon.com

**Bilal Fadlallah**
Amazon
bhf@amazon.com

## Abstract

Deep learning methods have achieved state-of-the-art performance in most modeling tasks involving images, text and audio, however, they typically underperform tree-based methods on tabular data. In this paper, we hypothesize that a significant contributor to this performance gap is the interaction between irregular target functions resulting from the heterogeneous nature of tabular feature spaces, and the well-known tendency of neural networks to learn smooth functions. Utilizing tools from spectral analysis, we show that functions described by tabular datasets often have high irregularity, and that they can be smoothed by transformations such as scaling and ranking in order to improve performance. However, because these transformations tend to lose information or negatively impact the loss landscape during optimization, they need to be rigorously fine-tuned for each feature to achieve performance gains. To address these problems, we propose introducing *frequency reduction* as an inductive bias. We realize this bias as a neural network layer that promotes learning *low-frequency representations* of the input features, allowing the network to operate in a space where the target function is more regular. Our proposed method introduces less computational complexity than a fully connected layer, while significantly improving neural network performance, and speeding up its convergence on 14 tabular datasets.

## 1 Introduction

Despite being one of the simplest and most natural ways to describe data, tabular data with heterogeneous columns poses a significant challenge for deep learning models. Recently, [5, 13] benchmarked various deep learning approaches against tree-based models and showed that tree-based models tend to perform the best on tabular data. Nevertheless, novel neural network architectures provide several advantages over classical machine learning methods. Specifically, neural networks operate on differentiable loss functions, are able to learn meaningful and high-dimensional data representations, and can scale to large datasets. In addition, the rapid advances of neural networks in generative image and language modeling clearly indicate their capability to encode complex information [8, 18]. In order to fully utilize these benefits on tasks that include tabular datasets, identifying and addressing the reasons behind neural networks' lack of performance on this form of data is essential.

Several influential research efforts on synthetic and image data point out that, despite being universal approximators, neural networks have a spectral bias, i.e., they tend to learn the low frequency components of a function much more easily than higher frequency components [22, 27]. In addition, recent empirical studies on tabular deep learning provide insights on why neural networks typically

---

[*]These authors contributed equally.

37th Conference on Neural Information Processing Systems (NeurIPS 2023).

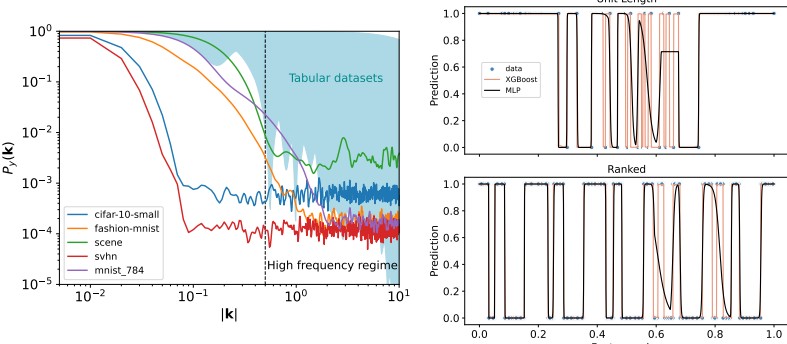

Figure 1: **(Left)** Due to their heterogeneous nature, tabular datasets tend to describe higher frequency target functions compared to images. The spectra corresponding to image datasets (curves in color) tend to feature lower Fourier amplitudes at higher frequencies than hetergoneous tabular datasets (cyan region). **(Top Right)** Data with labels varying over small distances are difficult for neural networks to learn, but easy for tree-based methods. **(Bottom Right)** Ranking is a transformation that redistributes the data and increases the scales over which the feature labels vary. This transformation effectively lowers the frequency of the target function, making it easier for a neural network to learn.

underperform on tabular data, including the latter's tendency to feature irregular target functions [13, 5]. Notably, the feature spaces of the tabular datasets considered in these studies are *heterogeneous*. Unlike images in tabular form where columns correspond to the same signal on different sensors, columns of tabular deep learning tasks typically represent features of different nature. These features are engineered to be informative as individuals (or subsets) and tend to have different statistical properties than each other.

In this work, we connect these lines of inquiry and claim that the spectral bias of neural networks may prevent them from learning the information present in the high frequency components of heterogeneous tabular datasets. We provide evidence to support this claim by analyzing the Fourier components of image and tabular datasets, as well as neural networks. To best of our knowledge, we present the first analysis on the relationship between spectral bias and tabular deep learning. We suggest that transforming tabular features into *low-frequency representations* may alleviate the negative impact of spectral bias. Notably, such transformations need to be performed in a way that balances the benefits of reduced frequency against potential information loss and additional negative effects on optimization (e.g., creating large gradients along certain directions) that can arise when altering the representation of the data. Figure 1 illustrates an example of the analyses we conduct. From the left panel, it can be observed that many tabular datasets tend to have higher power in their high frequency components, compared to some of the popular image datasets. We extend this comparison and present additional observations in Appendix C. The right panel depicts the impact of a frequency-reducing transformation on the functions learned by a multilayer perceptron (MLP) and Extreme Gradient Boosting (XGB) [7] on synthetic data. From this panel, it can be observed that the MLP is able to fit the target function better after the frequency-reducing transformation. Details of this analysis are provided in Appendix B.

Driven by these observations, we propose using *frequency reduction* as an inductive bias during neural network training. We apply this bias by introducing a trainable layer that promotes learning lower frequency feature mappings. We show that several commonly-used data processing methods impact the frequency spectrum of the target function as well as the resulting neural network learned from the data, however, their impact on performance varies with the dataset. Our proposal leads to the best performance, converges faster across all datasets considered, and introduces minimal computational complexity.

## 2   Related Work

**Spectral Analysis of Neural Networks.** Spectral properties of neural networks have been extensively studied. [22, 27] rely on Fourier analysis to investigate the learning behavior of neural networks,

observing that neural networks learn lower frequency components of target functions first. This phenomenon is referred to as *spectral bias* or the *frequency principle*. [3] study spectral bias on data with non-uniform density by examining the NTK for deep fully-connected networks, observing that both data density and frequency impact the gradient descent behavior. [6] show that during neural network training, error terms from different frequencies are controlled by the eigenvalues of the NTK. [16] decompose the reconstruction error of neural networks into low and high frequency components with respect to a cutoff frequency, and use these components to define a metric that quantifies spectral bias. [26] study the spectral bias of coordinate-based MLPs for modeling functions in low dimensions. Using NTKs, they show that mapping input coordinates to Fourier features [23] enables learning higher frequency information. Notably, they focus on the frequency described by pixels of a given image, rather than the frequency of the target function described by the whole dataset. The existing studies mainly focus on advancing the understanding of spectral bias on synthetic and image datasets, however, real-world tabular datasets are not considered.

**Tabular Deep Learning.** Due to neural networks' inferior performance on tabular data compared to tree-based models, various approaches have been proposed to close this performance gap. [1] take raw tabular data as input and use sequential attention to choose the set of features to use for every step of inference. This instance-wise feature selection is shown to improve performance and enable interpretability. [28, 2] extend self and semi-supervised learning to tabular data by introducing pretext tasks that aim to optimize a model to recover input samples from their corrupted variants. [25] leverage row- and column-wise attention combined with contrastive pre-training. [12] embed numerical features using linear transformations and categorical features using lookup tables, and feed these embeddings to a transformer. [15] show that using hyperparameter tuning, one can find a strong combination of 13 regularizers to be used with a simple MLP to outperform other tabular deep learning methods, as well as tree-based methods such as XGB. [11] propose individually embedding numerical features onto high-dimensional spaces before jointly using them in a model. The authors explore approaches such as piecewise linear embeddings that use bins selected by decision trees, and periodic activations to transform numerical features into high-dimensional representations.

**Discussion.** In most learning tasks that include images, audio or text, deep neural networks provide state-of-the-art performance. Breakthroughs in these domains are strongly connected to the design choices that exploit the salient characteristics of the corresponding data modality (e.g., convolutional layers in computer vision). On the other hand, existing studies on tabular deep learning mainly focus on applying approaches that are successful in other domains to improve performance [1, 12, 2, 25]. These studies do not rigorously investigate the fundamental reasons behind the performance gap between neural networks and tree-based methods. As a result, in order to gain performance, these approaches tend to sacrifice simplicity by complicating training and hyper-parameter tuning significantly. Such complexity causes these models to perform inconsistently across different studies, as utilizing the full potential of these models can be challenging. For example, [13] provide an extensive set of benchmarks, showing that tree-based approaches significantly outperform various tabular deep learning methods that originally claimed to outperform tree-based models. Similarly, [25] show that simple MLPs perform better than TabNet, an architecture that is tailored for tabular deep learning. As a result, instead of focusing on designing complex models to improve performance, we aim to identify and address the fundamental weaknesses of neural networks when learning from tabular data. We show that our approach improves performance of neural networks significantly without introducing additional complexity for fine-tuning, and helps them to converge faster.

## 3 Methodology

### 3.1 Background

Consider classification tasks defined on a tabular dataset $D = \{(\mathbf{x}_i, y_i)\}_{i=1}^{N}$ where $\mathbf{x}_i \in \mathbb{R}^n$ and $y_i \in \{0,1\}^m$ is the corresponding label. The task is to learn a classifier $f : \mathbb{R}^n \to \{0,1\}^m$. For simplicity, set $m = 1$ (i.e., binary classification), and let $y = 1$ correspond to the minority class in all cases. In Section 4, we provide experimental results and analyses for multi-class datasets as well, by extending this approach via one-vs-rest.

We are interested in the frequency spectra of various functions, including target functions implicitly defined by the data, for which $f(\mathbf{x}_i) = y_i$. Since we only have the labels defined at the datapoints supplied in each dataset, we rely on spectral analysis methods that take an arbitrary set of points as

input and output a measure of the strength of fluctuations at particular scales. One such method is a generalized Fourier transform known as the non-uniform discrete Fourier transform (NUDFT) [10]. For a function $f(\mathbf{x})$ and a set of points $X = \{\mathbf{x}_i\}_{i=1}^N$, the NUDFT defines a function of frequency vector $\mathbf{k} \in \mathbb{R}^n$ as

$$\widetilde{f}_X(\mathbf{k}) = \frac{1}{N} \sum_{i=1}^N f(\mathbf{x}_i) e^{-2\pi i \mathbf{k} \cdot \mathbf{x}_i}. \tag{1}$$

Equation (1) can be interpreted as an approximation of the multidimensional continuous Fourier transform (FT) of $f(\mathbf{x})$ weighted by the empirical distribution represented by the datapoints $X$, further discussed in Appendix A. [27] demonstrate the spectral bias of neural networks in terms of the NUDFT.

The Fourier transform approximated in Equation 1 evaluated at a given $\mathbf{k}$, corresponds to the coefficient of the $e^{2\pi i \mathbf{k} \cdot \mathbf{x}}$ term in a projection onto complex exponentials. Therefore, a natural definition of one function being *higher frequency* than another corresponds to the former having larger-magnitude Fourier amplitudes away from a specified low-frequency regime than the latter. We can make this comparison concrete by evaluating amplitudes along specific directions in $k$-space. Defining $\mathbf{k} = k\hat{\mathbf{k}}$, with $\hat{\mathbf{k}}$ a unit vector in $\mathbb{R}^n$, we can compare the size of the squared NUDFT amplitudes, $P_f(\mathbf{k}) = \left| \widetilde{f}_X(\mathbf{k}) \right|^2$, for functions along a given $\mathbf{k}$ direction by computing the sum

$$S_{\hat{\mathbf{k}}}[f] = \frac{2}{\overline{f}^2} \sum_{n=0}^{(k_* - k_0)/\Delta k} P_f\left( (k_0 + n\Delta k)\hat{\mathbf{k}} \right) \Delta k, \tag{2}$$

where $f$ is a function of interest. Here, $\overline{f}$ is the mean of $f(\mathbf{x})$ (i.e., $y$) evaluated on the the points in $X$, so that the scales at low frequencies are comparable for any given dataset (see Appendix A.2). Equation (2) can then be used to compare spectra: if $S_{\hat{\mathbf{k}}}[f_1] > S_{\hat{\mathbf{k}}}[f_2]$ for two functions $f_1$ and $f_2$ defined on $\mathbb{R}^n$, we say that $f_2$ is a *higher frequency function* than $f_1$ along the direction $\hat{\mathbf{k}}$, in the sense that it has more normalized signal energy (variance) in the region of interest, and refer to *frequency reduction* as reducing $S_{\hat{\mathbf{k}}}(\mathbf{k})$. In the above expression, $k_0$ defines the boundary between the low- and high-frequency regions, and $k_*$ is a cutoff frequency, discussed in Appendix A.4. Notably, Equation (2) only considers the magnitude of Fourier amplitudes along one dimension of $\mathbb{R}^n$ at a time. In Appendices A.3 and A.6, we discuss an alternative approach that works in the full feature space $\mathbb{R}^n$ and show that using (2) to evaluate along principal components (PCs) yields similar results to working in $\mathbb{R}^n$, while being more computationally efficient. Therefore, we rely on evaluations along PCs.

### 3.2 Spectral Analysis of a Class of Target Functions

The NUDFT, when evaluated using the labels $f(\mathbf{x}_i) = y_i$, can be understood in terms of the continuous Fourier transform of a class of target functions fitting the data. Given a binary classification task described by a dataset $D = \{(\mathbf{x}_i, y_i)\}_{i=1}^N$, a typical way to train a neural network $f$ is minimizing the empirical risk $1/N \sum_{i=1}^N \ell(y_i, f(\mathbf{x}_i))$, where $\ell$ denotes the cross-entropy loss. Assuming that $D$ maps each unique $\mathbf{x}_i$ to a single $y_i$, the cross-entropy loss without explicit regularization terms is exactly minimized for any function $f(\mathbf{x})$ such that $f(\mathbf{x}_i) = y_i \forall i \in \{1, \ldots, N\}$. This defines a set of equalities satisfied by any ideal target function perfectly fitting the training set[2], but does not further specify the form of the function away from the training instances. Indeed, there is no unique definition of the target function given only a finite set of points. However, we can parameterize a class of target functions minimizing the loss by convolving the target values at known data points with a given envelope function, $G(\mathbf{x}, \mathbf{x}')$, that falls off sufficiently rapidly away from the points $\{\mathbf{x}_i\}$:

$$\hat{f}[\mathbf{x}, G] \equiv \sum_{i=1}^N \int_{\mathbb{R}^n} y_i G(\mathbf{x}, \mathbf{x}'; \alpha_i) \delta^n(\mathbf{x}' - \mathbf{x}_i) d^n\mathbf{x}', \tag{3}$$

where $\delta^n(\mathbf{x})$ is the Dirac delta function in $\mathbb{R}^n$ and $\alpha_i$ denotes the parameters defining the envelope for the point $\mathbf{x}_i$. A typical loss function (neglecting regularization terms) will be exactly minimized

---

[2]In practice, some form of regularization will need to be utilized to avoid overfitting, but this set of equalities should still approximately hold. We verify empirically that the spectra of classifier functions learned in realistic settings approach the spectra of the target functions defined in this idealized sense.

provided $G(\mathbf{x}_i, \mathbf{x}_i; \alpha_i) = 1$ and $G(\mathbf{x}_i, \mathbf{x}_j; \alpha_i) = 0 \, \forall \, \mathbf{x}_i, \mathbf{x}_j \in D$. Perhaps the simplest such envelope is a multi-dimensional generalization of the boxcar function

$$B(\mathbf{x}', \mathbf{x}; \Delta_i) = \prod_{j=1}^{n} \mathrm{rect}\Big(\frac{x'_j - x_j}{\Delta_i}\Big), \tag{4}$$

where $\mathrm{rect}(\frac{a'-a}{\Delta}) = 1$ if $|a' - a| < \Delta/2$, $= 0.5$ if $|a' - a| = \Delta/2$, and vanishes otherwise. Notably, $\Delta$ corresponds to an upper bound on the $L_\infty$ distance between $a$ and $a'$, in order for them to be assigned to the same value of $y$. The target function defined by inserting Equation (4) into Equation (3) is then

$$f(\mathbf{x}) = \sum_{i=1}^{N} y_i \prod_{j=1}^{n} \int_{\mathbb{R}^n} \mathrm{rect}\left(\frac{x_j - x'_j}{\Delta_i}\right) \delta(x'_j - (\mathbf{x}_i)_j) d\mathbf{x}'. \tag{5}$$

Provided that $\Delta_i$ satisfies

$$|\mathbf{x}_i - \mathbf{x}_j|_\infty \geq \Delta_i \, \forall \mathbf{x}_j \in D, \tag{6}$$

for each $\mathbf{x}_i$, this function minimizes the cross-entropy loss on the training set, since its values at each point are simply the corresponding label. This corresponds to *tiling* the feature space with $n$-dimensional hypercubes of edge lengths $\Delta_i \forall i$, centered on each datapoint $\mathbf{x}_i$ and taking the target function to be constant within that region with value $y_i$. This is the simplest possible choice for the envelope function in that it does not assume any spectral features present in the target functions on distances smaller than $\Delta_i$.

We can now use the continuous Fourier transform to analyze the frequency spectrum of the target function defined in Equation (5). Denote the Fourier transform of a function $g(\mathbf{x})$ as $\mathcal{F}_\mathbf{x}[g(\mathbf{x})](\mathbf{k})$. By the convolution theorem, we have

$$\mathcal{F}_\mathbf{x}[f(\mathbf{x})](\mathbf{k}) = \sum_{i=1}^{N} y_i e^{-2\pi i \mathbf{k} \cdot \mathbf{x}_i} \Delta_i^n \left(\prod_{j=1}^{n} \mathrm{sinc}(k_j \Delta_i)\right), \tag{7}$$

where $\mathrm{sinc}(x) = \sin(\pi x)/(\pi x)$. Different choices for the $\Delta_i$ correspond to different assumptions about the target function away from the datapoints. One simple class of target functions is obtained by taking a common $\Delta_i = \Delta = \min\left\{|\mathbf{x}_i - \mathbf{x}_j)|_\infty\right\}_{i=1, j=i+1}^{N-1, N}$. This choice satisfies Eq. (6) and in the limit $|\mathbf{k}|_\infty \ll 1/(\pi \Delta)$, yields

$$\mathcal{F}_\mathbf{x}[f(\mathbf{x})](\mathbf{k}) \approx \Delta^n \sum_{i=1}^{N} y_i e^{-2\pi i \mathbf{k} \cdot \mathbf{x}_i} = N \Delta^n \widetilde{f}_X(\mathbf{k}) \tag{8}$$

where the approximation holds up to terms $\mathcal{O}(|\mathbf{k}|^2 \Delta^2 \pi^2)$ (here $||$ denotes Euclidean distance). The RHS above is the NUDFT up to a constant dimensionful scaling factor, which will drop out when using a procedure like Equation (2) that normalizes the amplitudes to the value at $k = 0$. Therefore, the NUDFT reproduces the spectral properties of a simple class of target functions defined by Equation (3) for frequencies below the scale set by the inverse minimum nearest-neighbor separation of points in the dataset. We implicitly enforce the latter restriction by cutting off our analysis at $k_*$, as discussed in Appendix A.4. Note that other choices for the $\Delta_i$ result in different target functions with spectra that can be analyzed by appropriately modifying the NUDFT.

In order to compare against the same quantity in all analyses, we also utilize Equation (1) for analyzing the frequency spectra of classifiers. A similar argument holds, and the corresponding results can be interpreted as analyzing the spectra of a classifier coarse-grained by convolving the neural network predictions at the datapoints with a narrow envelope as in Equation (5). In all cases, the classifier spectra defined in this way converge to that of the target function as training progresses, up to deviations caused by spectral bias (see also [27] for similar results). This suggests that high-frequency target functions, defined as above, are best fit by high-frequency classifiers, and that transformations affecting the target function spectrum will impact the frequency spectrum of the corresponding classifier.

| | house | electricity | phoneme | MagicTelescope | bankMarketing | MiniBooNE | albert | california |
|---|---|---|---|---|---|---|---|---|
| unit length | 1.31 | 1.46 | 1.66 | 0.13 | 2.01 | 16.24 | 1.74 | 1.5 |
| rank | 0.11 | 0.18 | 0.44 | 0.07 | 0.74 | 0.34 | 0.17 | 0.3 |
| scale | 0.01 | 0.18 | 0.06 | 0.01 | 0.17 | 0.35 | 0.04 | 0.03 |
| selrank | 0.11 | 0.22 | 0.58 | 0.07 | 1.28 | 0.34 | 0.29 | 0.32 |

| | house | electricity | phoneme | MagicTelescope | bankMarketing | MiniBooNE | albert | california |
|---|---|---|---|---|---|---|---|---|
| unit length | $88.94 \pm 0.17$ | $85.28 \pm 0.12$ | $87.78 \pm 0.28$ | $87.84 \pm 0.18$ | $\mathbf{90.56 \pm 0.11}$ | $90.78 \pm 0.04$ | $66.35 \pm 0.04$ | $88.51 \pm 0.15$ |
| rank | $89.61 \pm 0.14$ | $86.1 \pm 0.19$ | $88.0 \pm 0.46$ | $87.9 \pm 0.24$ | $90.41 \pm 0.08$ | $\mathbf{95.01 \pm 0.06}$ | $66.21 \pm 0.08$ | $88.51 \pm 0.17$ |
| scale | $89.65 \pm 0.16$ | $85.19 \pm 0.13$ | $\mathbf{88.64 \pm 0.21}$ | $87.88 \pm 0.23$ | $\mathbf{90.56 \pm 0.13}$ | $94.0 \pm 0.04$ | $\mathbf{66.48 \pm 0.05}$ | $\mathbf{88.53 \pm 0.22}$ |
| selrank | $\mathbf{89.91 \pm 0.16}$ | $\mathbf{87.04 \pm 0.17}$ | $88.15 \pm 0.32$ | $\mathbf{88.14 \pm 0.27}$ | $90.56 \pm 0.13$ | $94.94 \pm 0.05$ | $66.28 \pm 0.06$ | $88.48 \pm 0.2$ |

Table 1: (**Top**) Normalized sum of high-frequency Fourier amplitudes, $S_{\hat{\mathbf{k}}}$, for various binary classification datasets and transformations, averaged over the 1st principal component direction after each transform is applied. Ranking and scaling individual features tends to significantly reduce the high-frequency energy of tabular datasets relative to features normalized to unit scale. For an extended discussion, see Appendix A.6. (**Bottom**) Accuracy measurements corresponding to different transformations. Overall, lower frequencies are correlated with better performance. However, the benefits of frequency reduction is transformation- and dataset-dependent.

### 3.3 Case Study: Impact of Transformations on Target Function Spectra

Most neural networks need their input to be transformed in order to converge faster and generalize better. [14] show that these transformations serve to regulate layer input/output gradient statistics and/or the conditioning of input/gradient covariance matrices. A typical example of such transformations is scaling the features so that they vary over the unit interval $[0, 1]$, i.e., *min-max* scaling. Different choices of transformations can dramatically impact the target function frequency spectrum, and therefore the spectrum of the network that best fits the data with a given loss function. Due to the spectral bias of neural networks, we expect that using transformations that *reliably* reduce the high frequency components of the target functions result in better performance (see Appendix D for further discussion and illustration of this point).

Let us consider two types of such transformations, rank and scale, that are applied to each feature independently. Consider a vector $\mathbf{z}^\ell \in \mathbb{R}^N$ with elements $(\mathbf{z}^\ell)_r \in \{(\mathbf{x}_1)_\ell, \ldots, (\mathbf{x}_N)_\ell\}$ such that $(\mathbf{z}^\ell)_r \leq (\mathbf{z}^\ell)_s \, \forall r \leq s$. Then

$$\text{rank}((\mathbf{x}_i)_\ell) = \frac{1}{N}\min\left\{ r \in \mathbb{N} | (\mathbf{z}^\ell)_r = (\mathbf{x}_i)_\ell \right\}, \tag{9}$$

$$\text{scale}((\mathbf{x}_i)_\ell) = a \times (\mathbf{x}_i)_\ell, \tag{10}$$

for $\mathbf{x}_i \in X$ and $a \in \mathbb{R}$. Equation (9) simply maps a feature to the range $[0, 1]$, proportional to its position when ranked by magnitude along with the other values in the dataset. For any $(\mathbf{x})_\ell$ not in the original dataset, rank returns a value interpolated between $\text{rank}(\mathbf{x}_i)_\ell$ and $\text{rank}(\mathbf{x}_j)_\ell$ for the $(\mathbf{x}_i)_\ell, (\mathbf{x}_j)_\ell \in X$ nearest to $(\mathbf{x})_\ell$ such that $(\mathbf{x}_i)_\ell < (\mathbf{x})_\ell < (\mathbf{x}_j)_\ell$. Both scale and rank may alter the frequency spectrum of the target function described by a dataset by either stretching/squeezing fluctuations while preserving relative distances between points or neglecting distance information altogether and forcing the points to be uniformly distributed along a direction, respectively.

Using Equation (2), we present the high-frequency Fourier amplitudes (i.e., $|\mathbf{k}| > 0.5$) of the target functions of 8 tabular binary classification datasets with different transformations and their impact on performance across 10 random seeds in Table 1. Here, we implement scale as standardization, such that $a$ in Equation (10) is the inverse standard deviation of the feature in the training set population. From the top panel of Table 1, it can be observed that scale and rank consistently reduce the frequency of the target functions of all datasets, although by different relative amounts.

The bottom panel of Table 1 shows that, overall, lowering target function frequency is correlated with better NN performance: converting the averaged $S_{\hat{\mathbf{k}}}$ and NN performance for unit length, rank, and scale to their ranks within each dataset and comparing yields a Spearman's rank correlation coefficient of $\approx -0.7$ with a $p-$value of $\approx 3 \times 10^{-4}$. However, this correlation is not exact: large reduction in high-frequency Fourier amplitudes may not always result in significant performance improvements. This is because the amount of high frequency information encoded by each feature is different across datasets. Also, scaling up the feature values by a significant amount reduces the high-frequency energy, but can also negatively impact optimization by generating large gradients

during training. Finally, rank discards relative distance information between points in favor of frequency reduction, and also impacts optimization as it produces non-centered features. Therefore, it is important to simultaneously balance the effect of a given frequency-reducing transformation with other potentially negative impacts on the loss landscape or due to information loss.

These observations indicate that there is not a simple one-size-fits-all transformation that *reliably* and consistently reduces the high-frequency spectral energy that results in improved network performance. However, with this trade-off between frequency reduction and potential loss of information or effects on the loss landscape in mind, we can take a first step in improving over the rank and scale transformations by doing them *selectively*. For example, to balance the aforementioned trade-off we consider a selective version of rank defined as

$$\text{selrank}((\mathbf{x})_\ell) = \begin{cases} \text{rank}((\mathbf{x})_\ell), & S_{\hat{\mathbf{k}}_\ell}[\{\text{rank}((\mathbf{x}_i)_\ell), y_i\}_{i=1}^{N_{\text{tr}}}] \leq \Theta S_{\hat{\mathbf{k}}_\ell}[\{(\mathbf{x}_i)_\ell, y_i\}_{i=1}^{N_{\text{tr}}}] \\ (\mathbf{x})_\ell, & \text{otherwise} \end{cases} \tag{11}$$

where the datapoints in the conditions are those in the training set and $\Theta \in \mathbb{R}$. selrank applies rank only if it reduces the high frequency energy of the target function defined for the starting data representation by a certain relative amount, $\Theta$. Otherwise, selrank leaves a given feature as-is. The threshold $\Theta$ parameterizes how significant the frequency reduction must be in order to outweigh the potentially detrimental loss of information or impact on the loss landscape. From Table 1, it can be observed that selrank performs comparably or better than its counterpart rank across the datasets considered, motivating the use of frequency-informed transformations.

## 3.4 Frequency Reduction as Inductive Bias

Reducing the frequency of the target function may improve neural network performance by reducing the impact of spectral bias. On the other hand, the way we reduce frequency may impact other factors that play a significant role in neural network training. Since the interactions between transformations that reduce frequency and other learning dynamics are unique to how a feature is originally distributed, we can loosely formulate finding a transformation $g$ parameterized by $\theta_g$ that reliably and beneficially reduces the frequency as a constrained optimization problem:

$$\underset{\theta_f, \theta_g}{\text{argmin}} \quad \frac{1}{N} \sum_{i=1}^{N} \ell(y_i, f(g(\mathbf{x}_i; \theta_g)); \theta_f) \quad \text{s.t.} \quad \int_K P_{f \circ g}(\mathbf{k}) d\mathbf{k} < \int_K P_{f^*}(\mathbf{k}) d\mathbf{k}, \tag{12}$$

where $\theta_f$ denotes the trainable parameters of neural network $f$, $P$ can be evaluated on arbitrary (including unlabeled) points, $K$ denotes a high frequency region of $k$-space, and $f^*$ denotes a neural network that minimizes the empirical loss $1/N \sum_{i=1}^{N} \ell(y_i, f^*(\mathbf{x}_i; \theta_{f^*}))$. Equation (12) suggests that a beneficial frequency-reducing transformation can be found by limiting the space of acceptable solutions represented by $f$ to yield lower spectral energy in its high frequency components, compared to its analogous model $f^*$ trained on the raw data. One way to attempt solving this optimization problem is to simplify the frequency reduction constraint and directly use $P_f(\mathbf{k})$ to regularize the training loss towards finding a low frequency solution. However, this approach is computationally expensive, would in general require transformations on multiple features simultaneously, and would require careful pre-determination of the region $K$ to represent the spectral energy in high-frequency components of the learned function.

To alleviate these challenges we make the following simplifications. First, to avoid the computational complexity of working in the full feature space, we assume that all input features are conditionally independent given the target $y$, hence, they do not have interactions that significantly impact the frequency of the high dimensional decision boundary described by the data. Notably, we only use this assumption to limit our consideration to transformations that reduce the high frequency spectral energy by acting on individual features one at a time instead of jointly. After the transformation is conducted, consideration of feature interactions for the learning task is dictated by the underlying network architecture. Second, instead of relying on $S_{\hat{\mathbf{k}}}$ as a proxy for $\int_K P_f(\mathbf{k}) d\mathbf{k}$ to impose a frequency reduction constraint on $f \circ g$ during training, we constrain $g$ to be a composition of the two frequency-reducing transformations rank (9) and scale (10) (understood to be acting on the individual components of the feature vector $\mathbf{x}$):

$$g(\mathbf{x}; \theta_g : \{\theta_g^s, \theta_g^r, \theta_g^c\}) = \theta_g^c \, \text{scale}(\text{rank}(\mathbf{x}; \theta_g^r); \theta_g^s) + (1 - \theta_g^c) \, \text{scale}(\mathbf{x}; \theta_g^s) \quad \text{s.t.} \quad \theta_g^c \in [0, 1], \tag{13}$$

where $\theta_g^r$ represents the set of parameters specifying the mapping from original feature values to their ranked counterparts (e.g., how the interpolation is done), and $\theta_g^s$ denotes the set of coefficients

used to scale the input features. Note that $\theta_g^r$ is not a variable of this optimization problem as it only depends on the order of feature values in training set. Equation (13) describes a linear trajectory between scaled features and their ranked counterparts, while the degree of scaling applied to original and ranked features can be different. Notably, although we describe a linear relationship, one can encode more complex relationships to enable exploration of a larger space of frequency reducing transformations. With $g$ applying frequency reduction as a soft constraint, we drop the constraint from Equation (12), and set $\{\theta_f, \theta_g\}$ to the minimizers of the empirical loss. We implement Equation (13) as the input layer of a neural network and let $\theta_g$ get updated jointly with the rest of the network parameters during backpropagation. We satisfy the constraint $\theta_g^c \in [0, 1]$ by clamping these weights after each update. Note that, computational complexity introduced by Equation (13) is linear in the number of input features (see Appendix F).

It is important to note that $\mathrm{rank}$ and $\mathrm{scale}$ are used to individually reduce the high frequency spectral energy of the empirical distribution function $\hat{p}(\mathbf{x})$, even though our objective is to reduce the spectral energy for $f(\mathbf{x}; \theta_f) \sim P(y|\mathbf{x})$ instead. This is motivated by the observation that (13) is designed as an inductive bias that exploits the neural network $f$'s spectral bias, and $g$'s parameters are selected towards minimizing the empirical loss. Given that neural networks have strong spectral bias towards learning low frequency functions, when jointly optimized during training, $f$ prefers mapping the training instances to an informative low-frequency form using $g$ and fitting to this low-frequency alternative in order to minimize the loss further. In Section 4, we show empirical evidence that neural networks trained with $g$ indeed have reduced spectral energy in their high-frequency components compared to the networks trained on the same input data.

## 4   Experiments and Results

Our evaluation focuses on 3 key metrics: performance, rate of convergence and the irregularity of functions learned. Performance is evaluated using accuracy and the area under the receiver operating characteristic curve (AUROC). Rate of convergence is evaluated using the mean number of training epochs required to minimize validation loss. Finally, irregularity of functions learned by the neural network models are measured using total high-frequency power in Equation 2 along top principal components (PCs). We evaluate our proposed approach using 14 benchmark classification datasets listed in Table 2. These datasets are used by [13] to demonstrate the performance gap between tree-based models and neural networks. However, unlike [13], we use these datasets without truncating any samples, and we drop *id* and *date* type features as the approaches we compare are not designed to utilize the information conveyed by them. Because we consider a wide variety of datasets, the evaluation metrics we collect for the baselines and the proposed method highly fluctuate. This makes it challenging to draw conclusions for the overall behavior of these methods. Based on these observations, similar to [13], we normalize our measurements and aggregate them across datasets. In Appendix E, we provide the raw measurements and additional details on data preparation and training. Implementation details to reproduce our results are provided in Appendix H.

**Frequency Reduction with Neural Networks.** We present normalized and aggregated statistics across 14 datasets to highlight the general behavior of different transformations. Figure 2 depicts these statistics, where boxes represent quartiles, notches represent $95\%$ confidence intervals and whiskers represent the full range except outliers. From the figure, it can be observed that our proposed

| Name | #Samples | #Features | Source |
|---|---|---|---|
| electricity [9] | 45312 | 9 | https://openml.org/d/151 |
| house_16H | 22784 | 17 | https://openml.org/d/821 |
| pol | 15000 | 49 | https://openml.org/d/722 |
| kdd_ipums_la_97-small | 7019 | 61 | https://openml.org/d/993 |
| MagicTelescope [9] | 19020 | 11 | https://openml.org/d/1120 |
| bank-marketing [9] | 45211 | 17 | https://openml.org/d/1461 |
| phoneme | 5404 | 6 | https://openml.org/d/1489 |
| MiniBooNE [9] | 130064 | 51 | https://openml.org/d/41150 |
| eye_movements [24] | 10936 | 28 | https://openml.org/d/1044 |
| jannis | 83733 | 55 | https://openml.org/d/41168 |
| california [19] | 20640 | 8 | https://www.dcc.fc.up.pt/ltorgo/Regression/cal_housing.html |
| albert | 425240 | 79 | https://openml.org/d/41147 |
| credit card clients [9] | 30000 | 24 | https://archive.ics.uci.edu/ml/datasets/default+of+credit+card+clients |
| Diabetes [9] | 768 | 9 | https://www.openml.org/search?type=data&sort=runs&id=37 |

Table 2: 14 Tabular datasets used in the experiments

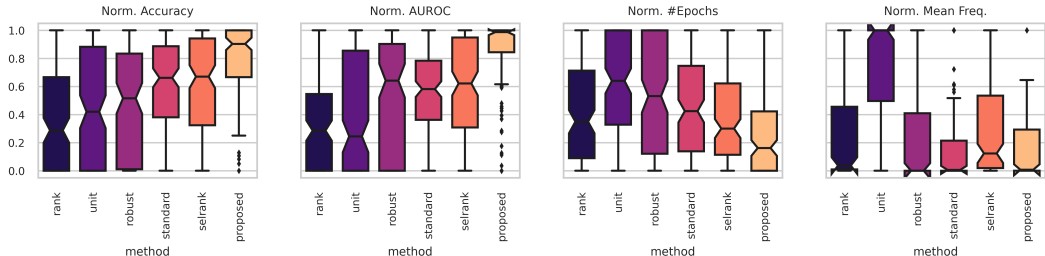

Figure 2: Normalized measurements of performance, convergence and frequency across datasets and seeds. The simple frequency-informed approach, selrank, with rank implemented as quantile transform improves upon rank alone. Additionally, our proposed methods significantly outperform all baselines, while converging faster.

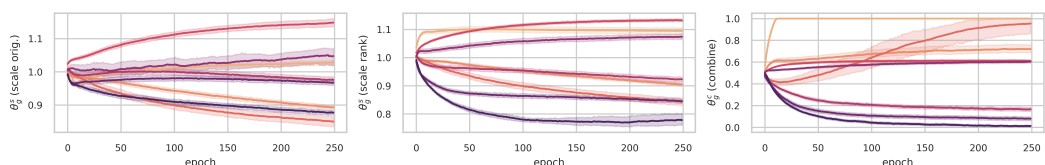

Figure 3: Weights learned by the proposed approach on *electricity* dataset. Lines represent values taken by the weights $\theta_g$ throughout 250 epochs, and shaded regions represent the 95% confidence intervals. The weights consistently converge to similar values across 10 random seeds that vary data splits and neural network initializations. Due to heterogeneous nature of tabular datasets, scaling and ranking weights that minimize the loss for different features are different.

method provides significantly higher normalized accuracy and AUROC, while taking significantly less epochs to converge. All of the transformations considered result in reduction of frequency of the function learned, compared to the neural network trained on unit scaled data. However, as discussed in Section 3.3, existing methods do not benefit from frequency reduction as much as our proposed methods, as they may have side effects such as losing relevant information (i.e., rank), or negatively impacting optimization (i.e., robust, standard and rank). On the other hand, our proposed method is able to balance the tradeoff between performance gain from frequency reduction and loss due to the side effects of the transformations used, providing an effective inductive bias for tabular deep learning.

**Investigating the Learned Weights.** Figure 3 depicts the change of the weights of our proposed approach across 10 random seeds on *electricity* dataset. Appendix E.1 extends this plot to all datasets. Across different seeds, initialization of our proposed layer's weights stay the same (i.e., $\theta_g^s := \mathbf{1}$ and $\theta_g^c := \mathbf{0.5}$), however, the network weights $\theta_f$ and data splits change. The first two columns of the figure correspond to the scaling weights $\theta_g^s$ for the raw features and ranked features, and the third column corresponds to the weights $\theta_g^c$ that combine these features from Equation (13). Shaded regions in the figure represent 95% confidence intervals. From the figure, it can be observed that the confidence intervals of the learned weights are small, relative to their change over epochs. This implies that the values of these weights stay consistent across random seeds. Consequently, the network consistently learns similar representations for the same feature and the representations do not overfit to the underlying network architecture. The figure also shows that the amount of scaling and ranking that can be applied to gain performance is data-dependent. Therefore, it is computationally expensive to exhaustively find the best transformations for each feature.

**Frequency Reduction with Tree-based Models.** XGB typically outperforms all other tabular learning approaches and is known to not be impacted by irregularity of target functions [13]. Therefore, in order to isolate the performance improvements observed by our approach to spectral bias of neural networks, it is important to confirm that our approach does not improve XGB performance. In Appendix E, we provide empirical evidence that our approach's impact to XGB is insignificant. Additionally, even though we do not intend this study to be a benchmarking of state-of-the-art methods, we see that using our approach, a simple MLP with no regularization and limited hyperparameter

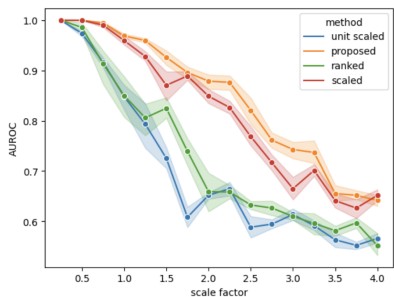

Figure 4: AUROCs of MLPs using the baseline transforms and the proposed method over multiple scale factors. As the scale factor increases, the target function frequency of the dataset also increases. Shaded regions correspond to $95\%$ confidence intervals across 10 random seeds.

tuning can outperform XGB in 4 of the datasets that were identified to be tree-friendly by previous studies [13], while providing competitive performance in 2 datasets.

**Frequency Reduction with other Network Architectures.** Frequency reduction as an inductive bias can improve the performance of more sophisticated deep learning architectures as well. For example, our proposed layer's mappings can be used as an input to TabNet to significantly reduce the amount of hyperparameter tuning required. As discussed in Section 2, it is challenging to utilize TabNet's full potential due to its hyperparameter sensitivity, causing inconsistent results across different studies. However, this model also offers additional benefits beyond performance improvements such as interpretability. Therefore, we believe TabNet is a good candidate to evaluate our inductive bias on beyond the vanilla MLP. Similar to our other experiments, we limit the number of hyperparameter configurations swept for tuning to 100. For each dataset, we train an MLP with our proposed layer until convergence and use the layer to generate low-frequency versions of the corresponding dataset. We compare the performance of our approach to TabNet trained by following the author's suggestions [1]. We find that our approach improves TabNet's AUROC by $4\% \pm 1.6\%$ (see Appendix E.2).

**Performance over Varying Target Function Frequency.** Due to the diverse nature of the datasets considered in our experiments, it may appear challenging to reliably draw conclusions about the effect of the proposed method. Different dataset characteristics naturally lead to variation in the performance gain observed from our methods. To address these concerns, we provide additional results using synthetic data. Specifically, we generate synthetic datasets of varying target function frequency by applying a common scale factor to $k_i$ in Equation (24) while keeping all other parameters fixed. We train the same 2-hidden-layer MLP for 200 epochs on the unit-scaled, ranked, and scaled (via standardization) features. Figure 4 depicts the AUROCs of these MLPs over different target function frequencies. From the figure we observe that for low target function frequencies, all methods perform comparably. Also, performance decreases with increasing target function frequency due to NN's spectral bias. Our proposed method consistently outperforms the others and is the most robust to high frequency target functions.

## 5   Conclusion

In this paper, we studied the impact of spectral bias of neural networks on tabular datasets. We showed that tabular datasets tend to describe irregular target functions and this irregularity negatively impacts neural network performance. We proposed incorporating frequency reduction as an inductive bias when training neural networks to reduce the irregularity of these functions. We showed that our proposed method significantly improves neural network performance while reducing the number of training epochs needed to converge. Although we propose the new direction of using frequency reduction as an inductive bias for tabular deep learning, we merely scratched the surface when it comes to realizing said bias. For example, our analyses can be extended to other tabular deep learning approaches to determine if the performance improvements they offer can be explained from the lens of spectral analysis. We discuss other future directions as well as the limitations of our work in Appendix G.

## Acknowledgements

We thank Tian Tan, Xuan Li, Man-Ho Marco Tang, Mihir Pendse and Jiajian Lu for valuable discussions and feedback. We are also grateful to the anonymous reviewers for providing constructive feedback on this manuscript.

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

## Supplementary Material

## A    Fourier Analysis Details

In this section, we provide additional details regarding our Fourier analysis.

### A.1    The NUDFT and its continuous limit

The primary starting point for our spectral analysis is the non-uniform discrete Fourier transform defined in Equation (1). This particular transform conveys spectral information about functions sampled at arbitrary data points and its adjoint is commonly used in the medical imaging domain to reconstruct images from measurements taken in Fourier space. Equation (1) can be interpreted as an approximation of the multidimensional continuous Fourier transform (FT) of $f(\mathbf{x})$ weighted by the empirical distribution represented by the datapoints $X$. Its continuous limit as $N \to \infty$ is the Fourier integral

$$\mathcal{F}_{\mathbf{x}}[f(\mathbf{x}); \mu](\mathbf{k}) = \int_{\mathbf{R}^n} f(\mathbf{x}) e^{-2\pi \mathbf{k} \cdot \mathbf{x}} \mu(d\mathbf{x}), \qquad (14)$$

where $\mu(d\mathbf{x})$ is the probability measure representing the distribution from which the points in the dataset are sampled.

The NUDFT (as well as any spectral analysis method relying solely on information at a finite number of points without additional interpolation or assumptions) reflects the spectral information of not only the target function, but also the underlying distribution of the datapoints itself. It is in principle possible to separate out these contributions using density estimation methods, e.g., a multidimensional generalization of the technique in [16]. Presently, however, we simply define the frequency spectra of interest as that reflected in Equation 14 (and its approximations) and frame our analysis self-consistently in terms of this definition. In particular, we empirically observe spectral bias (see also [27]), see that certain transformation reduce high frequency energy, and propose an inductive bias that consistently outperforms other methods, all using this definition.

### A.2    Limiting behavior of the NUDFT with respect to $|\mathbf{k}|$

Some of the limiting behavior of the NUDFT amplitude spectrum can be understood from Equation (1). Specifically, consider the expression for $P_f(\mathbf{k})$:

$$P_f(\mathbf{k}) = \frac{1}{N^2} \left| \sum_{i=1}^{N} f(\mathbf{x}_i) e^{-2\pi i \mathbf{k} \cdot \mathbf{x}_i} \right|^2 = \frac{1}{N} \overline{f^2} + \frac{2}{N^2} \sum_{i=1}^{N-1} \sum_{j=i+1}^{N} f(\mathbf{x}_i) f(\mathbf{x}_j) \cos\left(2\pi \mathbf{k} \cdot (\mathbf{x}_i - \mathbf{x}_j)\right).$$
$$(15)$$

The overline represents the mean of a quantity evaluated over the $\mathbf{x}_i$. At small $|\mathbf{k}|$, the RHS of Equation (15) is approximately $\overline{f}^2$, the mean squared of the function or labels. This is the reason for the $1/\overline{f}^2$ scaling in front of Equation (2); otherwise, the scales of the Fourier amplitudes would not be directly comparable across datasets. As $|\mathbf{k}|$ increases from 0, at some point, $P_f$ starts to deviate from the mean squared. This happens when the *cos* term in the expression above begins to deviate from 1 for a significant number of points. This occurs schematically when $|\mathbf{k}| \sim 1/(2\pi\Delta x)$, where $\Delta x$ is a *typical* separation between datapoints. In the other limit, once $|\mathbf{k}| \gg 1/(2\pi\Delta x)$, the NUDFT is probing distances much smaller than the separations of points in the dataset. At this point, the NUDFT ceases to provide meaningful spectral information, as the dataset does not convey information about the target function on such small scales. The spectrum becomes noisy, since the last term on the RHS of Equation (15) features many cancellations, resulting in fluctuations around the first term, $\overline{f^2}/N$, on the RHS. The most interesting region from the standpoint of our analysis is that between these two regimes, describing collective oscillatory behavior, which is the region we focus on in our analysis.

There is another way to see that the $1/N$ asymptotic behavior of $P_f$ at large frequencies is an artifact of the discrete distribution of datapoints. The NUDFT approximates the integral $\int_{-\infty}^{\infty} f(\mathbf{x}) \rho(\mathbf{x}) e^{-2\pi i \mathbf{k} \cdot \mathbf{x}} d\mathbf{x}$. For $f(\mathbf{x})\rho(\mathbf{x})$ continuous, bounded, and compactly supported, the integrated spectral energy density will be finite. However, the corresponding integral of the squared

NUDFT amplitudes, which should yield the same result as $N \to \infty$ by the Plancherel theorem, does not exist for finite $N$. The problem again arises from the behavior discussed in the previous paragraph. To obey the Plancherel theorem, the result in the continuous limit needs to fall off fast enough in $|\mathbf{k}|$ to be integrable, meaning that the $\overline{f^2}/N$ asymptotic behavior discussed above is an artifact of the discrete approximation. We therefore limit our analysis to below this range, which motivates the upper cutoff $k_*$ appearing in several of our expressions.

## A.3 Approximating the high-dimensional spectral energy integral

Our spectral analysis primarily relies on the quantity $S_{\hat{\mathbf{k}}}$ defined as a sum of squared NUDFT amplitudes as in Equation (2). This method considers spectral energy along single dimensions of $\mathbb{R}^n$ at a time. Here we describe an alternative *convolution method* for approximating the full $n$-dimensional spectral energy integral over a finite high-frequency region in $\mathbb{R}^n$. In Appendix A.6 we show that both methods yield similar results for the datasets analyzed.

The method here is similar to that in [16]. Consider the approximate energy spectral density, $P_f(\mathbf{k})$, for a function $f$ integrated over a high-frequency region $K \subset \mathbb{R}^n$,

$$S_K = \int_K P_f(\mathbf{k})d\mathbf{k}. \tag{16}$$

Rather than attempting to perform this high-dimensional integral in Fourier space, we can make use of the convolution theorem and work directly in feature space. Consider a region $K$ defined by $k_0 < |\mathbf{k}|_\infty \leq k_*$, where $||_\infty$ is the $L_\infty$ norm, $k_0$ is a threshold determining the start of the high-frequency region, and $k_*$ is a cutoff of the high frequency region introduced for the same reason as discussed in the previous section. We can then write

$$S_K = \int_{\mathbb{R}^n} P_f(\mathbf{k}) \mathbb{1}_{|\mathbf{k}|_\infty \leq k_*} d\mathbf{k} - \int_{\mathbb{R}^n} P_f(\mathbf{k}) \mathbb{1}_{|\mathbf{k}|_\infty \leq k_0} d\mathbf{k}. \tag{17}$$

Defining the function

$$g(\mathbf{x}) = \int_{\mathbb{R}^n} f(\mathbf{x}')(2k_*)^n \prod_{i=1}^n \text{sinc}\left(2k_*(\mathbf{x} - \mathbf{x}')_i\right) \mu(d\mathbf{x}'), \tag{18}$$

the convolution theorem dictates that

$$\mathcal{F}_{\mathbf{x}}[g(\mathbf{x}); \mu](\mathbf{k}) = \mathcal{F}_{\mathbf{x}}[f(\mathbf{x}); \mu](\mathbf{k}) \, \mathbb{1}_{|\mathbf{k}|_\infty \leq k_*}, \tag{19}$$

so that

$$\int_{\mathbb{R}^n} |g(\mathbf{x})|^2 \, d\mathbf{x} = \int_{\mathbb{R}^n} |\mathcal{F}_{\mathbf{x}}[g(\mathbf{x}); \mu](\mathbf{k})|^2 \, d\mathbf{k} = \int_{|\mathbf{k}|_\infty \leq k_*} P_f(\mathbf{k})d\mathbf{k}. \tag{20}$$

Approximating the $\mathbf{x}'$ integral in Equation (18) with a Monte Carlo integral sampling from the density defined by $\mu$, substituting the resulting expression into Equation (20) and using the definition in Equation (17), we arrive at

$$S_K[f] \approx \frac{(2k_*)^n}{N^2} \sum_{i,j=1}^N f(\mathbf{x}_i)f(\mathbf{x}_j) \left(\prod_{\ell=1}^n \text{sinc}\left[2k_*(\mathbf{x}_i - \mathbf{x}_j)_\ell\right] - \prod_{\ell=1}^n \text{sinc}\left[2k_0(\mathbf{x}_i - \mathbf{x}_j)_\ell\right]\right), \tag{21}$$

This estimate of the spectral integral in Equation (16) is evaluated for binary classification datasets in Appendix A.6 below.

## A.4 Defining the high-frequency regime

Both the NUDFT and fully-integrated energy spectrum approach rely on two different cutoffs to define the high-frequency range. The lower cutoff, $k_0$, sets the start of what we consider the high-frequency range. We take $k_0 = 0.5$ in our analyses, motivated by two observations. First, in our conventions, and for features scaled to unit length, $|\mathbf{k}| < 0.5$ corresponds to frequency modes that fit less than half a period in any given dimension of the feature space. This corresponds to long-distance (i.e., global) fluctuations and so the high-frequency regime should be considered for $|\mathbf{k}|$ larger than this value. The second motivation comes from the behavior of the normalized sinc function, which describes

the Fourier transform of the unit rectangle function. Say we had a dataset with all points having the same label distributed uniformly in the unit interval. The NUDFT would then approximate the normalized sinc function, which begins to decay when its argument is around 0.5. For larger $|\mathbf{k}|$, the envelope of the squared NUDFT amplitudes will decay as $1/|\mathbf{k}|^2$, consistent with the fact that the corresponding target function does not have high-frequency components. Of course we are also free to choose values larger than 0.5. We have verified that the other choices of $k_0 \sim 1$ do not affect our conclusions.

As pointed out in Appendix A.2, at high frequencies the behavior of our methods relying on discrete points fail to reproduce the correct decaying high-frequency behavior of the continuous Fourier amplitudes. Care must therefore be taken to properly select $k_*$, especially for computing $S_K$: for $S_{\hat{\mathbf{k}}}$ the high-frequency $\overline{f^2}/N$ behavior adds a constant, but this behavior leads to $S_K \propto (2k_*)^n$, which would dominate the result numerically if we did not account for this behavior. As a result we implement the following procedure in computing $S_K$ to compare between datasets: for each dataset, we first compute $S_K$ for several different values of $k_*$, and choose a value near where the integral begins to scale as $(2k_*)^n$. We then manually enforce that the contributions to $S_K$ above this value vanish. The corresponding integral then has the expected behavior, and we can directly compare $S_K$ between datasets at arbitrarily large $k_*$, since the integral will saturate once it comes close to scaling as $(2k_*)^n$. Finally, for computing $S_{\hat{\mathbf{k}}}$, we simply cut off the sum in Equation (2) at a given value near where $S_{\hat{\mathbf{k}}}$ approaches the $\overline{f^2}/N$. In practice, we take $k_* = 20$ when computing $S_{\hat{\mathbf{k}}}$, but have verified that the results do not significantly change when varying this value.

## A.5 Evaluating Fourier transforms along principal components

In several of our analyses, we evaluate the NUDFT $\widetilde{f}_X(\mathbf{k})$ along specific directions, such as those defined by the principal components (PCs) of the data. This approach was also used in [27] and we provide further details about this procedure here for evaluating along general directions in $\mathbb{R}^n$

The principal component decomposition (i.e., PCA) typically takes one point $\mathbf{x}$ in the original feature space and yields a transformed vector $\mathbf{x}_{\mathrm{PC}}$ with entries corresponding to the projection of the mean-centered features onto the PC directions. Focusing on one PC direction, we have

$$x_{\mathrm{PC}} = \hat{\mathbf{e}}_{\mathrm{PC}} \cdot (\mathbf{x} - \mathbf{m}), \tag{22}$$

where $\mathbf{m}$ is the vector defined by the mean of each $n$ dimensions of the feature space over the dataset, and $\hat{\mathbf{e}}_{\mathrm{PC}}$ is a unit vector pointing along the PC of interest in the original feature space. To take the Fourier transform along the direction $\hat{\mathbf{e}}_{\mathrm{PC}}$, we can define $\mathbf{k}_{\mathrm{PC}} \equiv k_{\mathrm{PC}}\hat{\mathbf{k}}_{\mathrm{PC}}$, where $k_{\mathrm{PC}} = \mathbf{k} \cdot \hat{\mathbf{e}}_{\mathrm{PC}}$ and $\hat{\mathbf{k}}_{\mathrm{PC}} = \hat{\mathbf{e}}_{\mathrm{PC}}$, and plug this value of $\mathbf{k}_{\mathrm{PC}}$ into Equation 1. Using Equation 22, we have

$$\sum_{j=1}^{N} f(\mathbf{x}_j)e^{-2\pi i \mathbf{k}_{\mathrm{PC}} \cdot \mathbf{x}_j} = e^{-2\pi i \mathbf{k}_{\mathrm{PC}} \cdot \mathbf{m}} \sum_{j=1}^{N} f(\mathbf{x}_j)e^{-2\pi i k_{\mathrm{PC}} x_{\mathrm{PC},j}}. \tag{23}$$

The phase factor in front of the RHS will drop out when we take amplitudes. The above expression tells us that to obtain the FT along the direction $\hat{\mathbf{e}}_{\mathrm{PC}}$ defined by a given principal component, we can simply plug in the values of $x_{\mathrm{PC}}$ corresponding to a given datapoint and treat the Fourier transform as one-dimensional with frequency $k_{\mathrm{PC}}$ (up to an overall phase factor). However, this is *not* a one-dimensional Fourier transform: it is the full (approximate) $n$-dimensional Fourier transform, *evaluated* along a given one-dimensional subset of $\mathbb{R}^n$.

The above method applies when evaluating a dataset along one of its own PCs. In computing $S_{\hat{\mathbf{k}}}$, e.g., in Table 1, we typically average over the PCs defined for different transformations. In this case, one should directly substitute the corresponding $\hat{\mathbf{k}}_{\mathrm{PC}} := \hat{\mathbf{e}}_{\mathrm{PC}}$ into the LHS of Equation (23).

## A.6 Spectra of binary classification datasets and correlation between $S_{\hat{\mathbf{k}}}$ and $S_K$

Here, we provide supplementary details related to our spectral analysis of the various binary classification datasets considered in our experiments.

In Figure 5 we show squared NUDFT amplitudes for the binary classification datasets of interest. The squared amplitudes $P_y(\mathbf{k})$ are evaluated for unit-scale, ranked and scaled data, with rank implemented as a quantile transform and scale as standardization. The amplitudes are evaluated

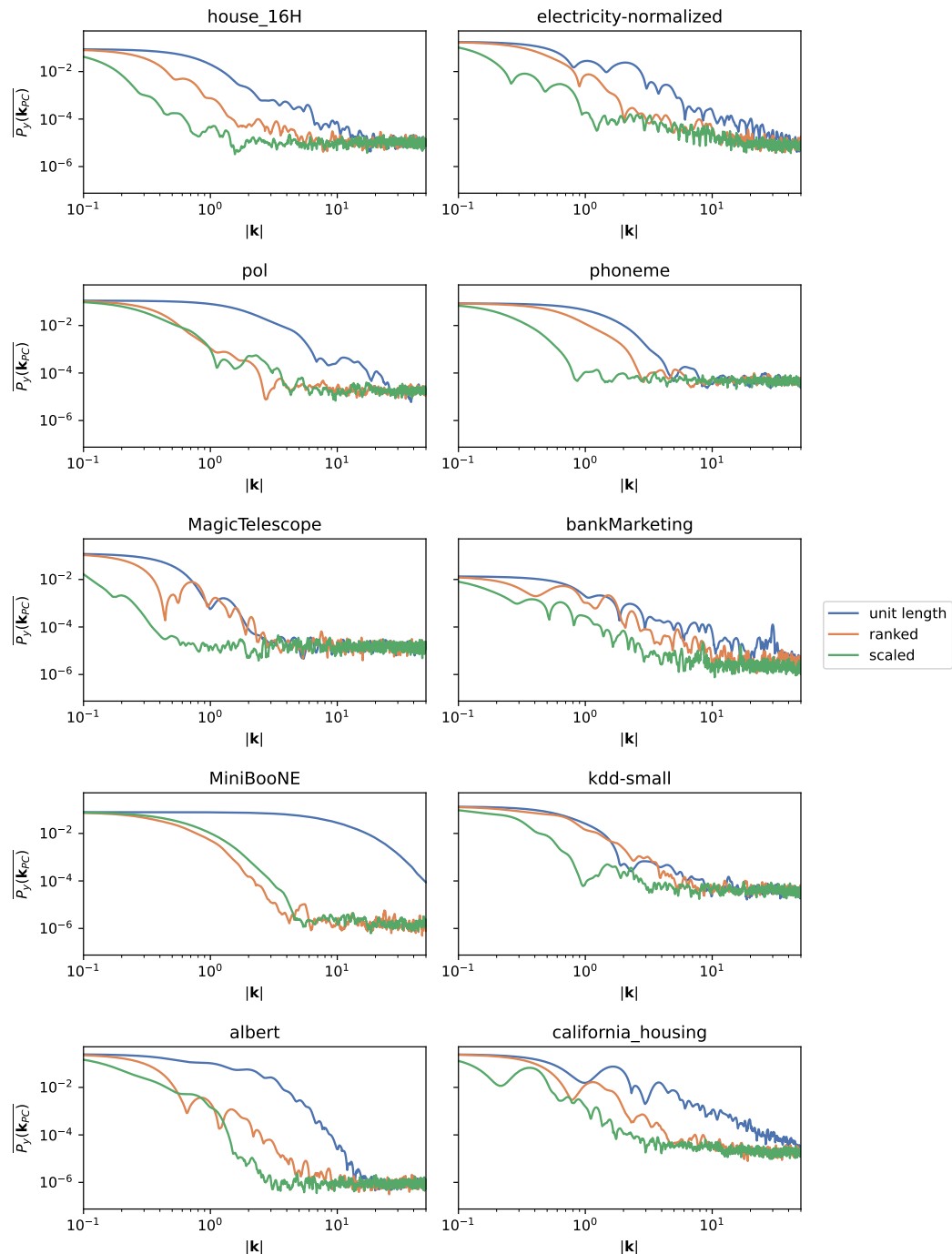

Figure 5: Spectra for binary classification training datasets determined by Equation (1) for the rank and scale transformations, compared to the datasets with unit length features. Results are marginalized over 10 different random seeds for the train-test split, and over $\hat{\mathbf{k}}_{\mathrm{PC}}$ corresponding to the 1st PC after each transformation is applied. The solid lines are the resulting mean values, with shaded regions representing the 95% confidence intervals. Different transformations significantly affect target function frequency spectra.

along the first PC after each transformation and for 10 different random training splits of the data. Solid curves represent the mean and shaded regions the corresponding 95% confidence intervals after

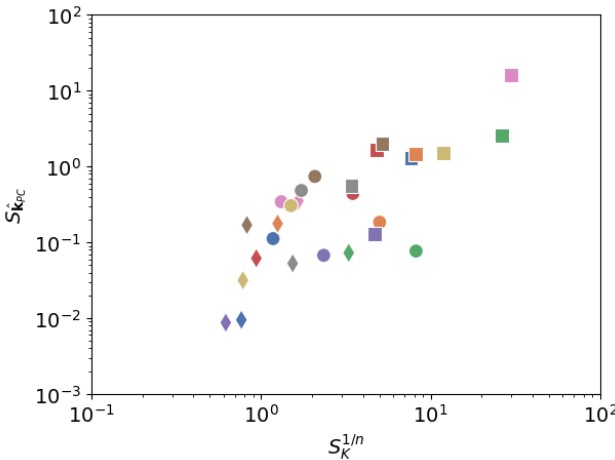

Figure 6: Comparison of different methods for quantifying the spectral energy at high frequencies. The horizontal axis corresponds to Equation (21), and the vertical axis to Equation (2). Squares, circles, and diamonds correspond to unit-scaled, rank, and scale features, and the different colors correspond to different datasets. The two methods are highly correlated.

marginalizing over all random seeds and different $\hat{\mathbf{k}}_{\mathrm{PC}}$ directions. From the figure, it can be observed that the various transformations affect the high-frequency composition of these datasets in different ways, and in some more than others.

The corresponding values of $S_{\hat{\mathbf{k}}}$ for several of these datasets are presented in Table 1. These are computed in the same way as the results in Figure 5 (i.e.,averaged over random seeds and PCs), however still correspond to examining the spectrum along one dimension at a time. To assess the extent to which the one-dimensional results represent the full spectral properties in $\mathbb{R}^n$, in Figure 6, we compare the resulting values of $S_{\hat{\mathbf{k}}}$ against the quantity $S_K$ of Appendix A.3. Results are shown for all binary classification datasasets except for *albert*, which features $k_*$ below $k_0$ for some methods when computed using the full Fourier integral. In order to compare the different datasets at similar scales, we compute $S_K^{1/n}$, since the dimensions of the volume integral over $\mathbf{k}$ will be different for datasets with different feature space dimension. In computing $S_K$, due to the $\mathcal{O}(N^2)$ computational complexity, we subsample each dataset to 10000 records if larger than this size. As seen in Figure 6, the two methods are strongly correlated: each dataset individually has a Spearman rank correlation coefficient of 1 when comparing $S_{\hat{\mathbf{k}}}$ and $S_K$. From the figure, we also see the general behavior demonstrated in the spectral plots previously: the unit-scaled representation tends to imply the most high-frequency target functions, followed by rank, implemented as a quantile transform, and scale, implemented as standardization.

## B    Spectral Analysis of Neural Networks on Synthetic Data

In this section, we provide additional details and experimental results related to the synthetic data experiments reflected on the RHS of Figure 1 (reproduced in Figure 7 for convenience).

We consider binary classification on a synthetic dataset consisting of a single numerical feature, $x$, for simplicity. Heterogeneous tabular datasets used for classification tend to feature rapid variations of the class labels over short distances in feature space. To simulate this scenario, consider a normally-distributed feature $x$, and an oscillatory target function that depends on a function $g(x)$ of the feature $x$, generating the class label as

$$y(x) = \left\lfloor \frac{1}{2n} \sum_{i=1}^{n} \left(1 + A_i \sin\left(k_i g(x) + \varphi_i\right)\right) + \frac{1}{2} \right\rfloor. \tag{24}$$

When sufficiently high frequency components $k_i$ are considered, a simple multilayer perceptron (MLP) will have difficulty learning the target function, as expected from the spectral bias phenomenon. This is illustrated in the top panel of Figure 7. For this example, we take $x \sim \mathcal{N}(0.5, 0.01)$,

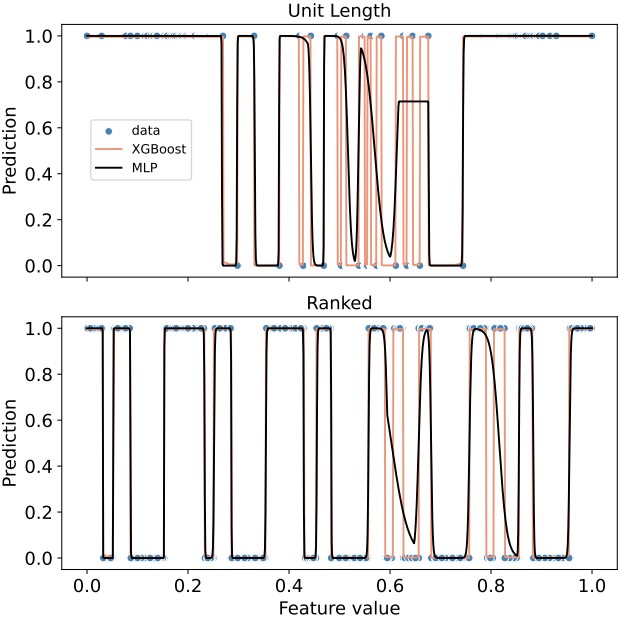

Figure 7: (Top) Data with labels varying over small distances are difficult for neural networks to learn, but easy for tree-based methods. (Bottom) Transformations that redistribute the data can increase the scales over which the feature labels vary, effectively lowering the frequency of the target function and making it easier for a neural network to learn.

$k_i \in \{1, 5, 10, 20\} \times 2\pi$ and randomly sample the corresponding amplitudes $A_i$ and phases $\varphi_i/2\pi$ from the continuous uniform distribution on $[0, 1]$. We also take $g(x)$ as the rank transformation of Equation (9), for illustration. In order to compare NN's with features at a common scale, we first min-max normalize the features to have unit length. The corresponding target function has $S_{\hat{\mathbf{k}}} \approx 0.02$. A 2-hidden-layer MLP with width 256 is then trained to classify 5,000 points using binary cross-entropy loss for 10000 epochs after an initial 250 epoch tuning stage to select the best learning rate. The resulting predictions are shown in black on the top panel of Figure 7, and do not match the training data very well, particularly in regions where the tag varies more rapidly (i.e., the spectral bias phenomenon). The NN evaluated on a validation set of 1000 points distributed in the same way as the training set yields an AUROC of 0.935. On the other hand, we also train an XGBoost classifier using default hyperparameters for 100 boosting iterations (the performance converges quickly so boosting for additional iterations does not improve performance). The corresponding predictions are shown in orange. This tree-based classifier performs well and is able to reproduce the rapid variations of the target function, since, as long as the algorithm's histogram binning is fine enough, it is straightforward to recursively split this one-dimensional feature space into regions with $y = 0$ and 1. The corresponding AUROC is 0.998.

There can be several approaches taken to overcome this performance gap. One approach is to focus primarily on the underlying model: perhaps tuning the hyperparameters extensively or incorporating a more sophisticated architecture could lead to a better-performing configuration. However, this process quickly becomes computationally intensive and is dataset-dependent. An alternative approach, and the one we study in this work, is to focus on the representation of the data. The frequency characteristics of a function defined on a dataset are inherently tied to the representation of the data itself. Stretching or warping the feature space while maintaining the label values will in general change the length scales over which the function varies.

To investigate this effect, consider the impact of re-arranging the datapoints such that rather than following their original distribution, they are now spaced as far apart from each other as possible within a fixed interval, which we take to be $[0, 1]$. This is a familiar transformation: mapping the original distribution to a uniform distribution along the $x$ feature direction, as in the rank transformation of Equation (9). However, the resulting effect on the example in Figure 7 is dramatic: this re-distribution of points reduces the frequency of the target function, as clearly seen in the bottom

panel (the rank-transformed target function has $S_{\hat{\mathbf{k}}} \approx 0.01$). We train the same two models (MLP and XGBoost) with the same hyperparameters as before on the transformed data. The tree-based model performance does not change significantly, but the MLP's performance improved dramatically, with the AUROC evaluated on the transformed validation set increasing from 0.935 to 0.989.

For this synthetic example, the simple inductive bias of seeking a representation of the data that "smooths out" the target function in a way that does not negatively effect optimization significantly boosts neural network performance and reduces the performance gap with respect to the GBDT-based model. Our analyses utilizing realistic heterogeneous tabular datasets yield similar results.

### B.1 Connection to *computable spectral bias*

Since we know the underlying density of the synthetic datapoints, we can directly compute the spectral bias of the MLPs trained on the pre- and post-transformed data following the computable definition of spectral bias proposed by [16]. In particular, a quantitative definition of the spectral bias, $SB$, can be provided in terms of the fraction of variance unexplained (FVU) compared between a predefined low- and high-frequency region. A value of $SB > 0$ indicates a larger fraction of FVU in the high-frequency region, hence a spectrally-biased NN, since the NN fits the data worse at high frequencies. To compare our two NNs, we consider an epoch for which the performance of the two NNs is comparable: the epoch with largest validation AUC for the unit length NN and the epoch with the closest corresponding performance for the ranked NN. The FVUs are approximately equal for both networks, however the spectral bias of the ranked NN, computed as in [16], is lower than that of the NN trained on unit-length features: $SB = 0.58$ for unit length, $SB = 0.42$ for rank. This indicates that for NNs achieving similar performance, the rank transformation has produced a network with a lower fraction of its error in high frequencies, and therefore has reduced the negative impact of spectral bias on NN training, since the relatively larger low-frequency errors can be effectively reduced by further epochs of training, in contrast to high-frequency errors, which are more difficult for the NN to reduce. This allows for significantly better performance overall for the rank-transformed data. Similar effects are suggested by our analysis of realistic multidimensional tabular datasets: transformations resulting in lower-frequency NNs for a fixed performance metric can typically be further trained to reduce their correspondingly lower-frequency errors, hence mitigating the impact of spectral bias in learning the corresponding target function. Notably, this analysis is an attempt to establish the initial connection between our approach and [16]. A more extensive analysis of the *computable spectral bias* of all the methods we consider in this paper will be included in future work.

## C   Frequency of Target Functions in Tabular and Image Datasets

In this section, we provide details related to the results on the LHS of Figure 1, contrasting the target function spectra of image vs tabular datasets.

Our study focuses on mitigating the effects of spectral bias in the context of tabular deep learning. One of the empirical motivations for our approach is the observation that target function spectra of tabular datasets tend to be of higher-frequency than similar examples from popular image datasets. To compare frequencies across datasets, the features should be normalized in a consistent way. We take unit length normalization as the baseline for our comparisons. We consider several popular benchmark datasets from computer vision applications: *cifar-10-small* (https://openml.org/d/40926), *fashion-mnist* (https://openml.org/d/40996), *mnist-784* (https://openml.org/d/554), *svhn* (https://openml.org/d/41081), and *scene* (https://openml.org/d/41471). After min-max normalizing, we treat these image datasets in the same way as our tabular multi-class datasets: for each of 10 random seeds, we consider a training set sampled from the full dataset, and compute the NUDFT along the first PC direction, assigning binary labels for all 1-vs-rest class assignments, and computing the mean over all seeds and binary label assignments. The resulting squared NUDFT amplitudes, $P_y(\mathbf{k})$, are plotted in color on the LHS of Figure 1. To compare against tabular dataset spectra, we use the same procedure. The envelope containing all such spectra computed for our 14 tabular benchmarks is shaded in blue on the LHS of Figure 1. In this figure, all amplitudes shown are normalized with an implicit $1/\overline{f}^2$, to account for the different levels of class imbalance across the datasets.

From these results, we immediately see that, using our definitions, the image datasets considered have significantly smaller high-frequency Fourier amplitudes when all features are normalized to the same scale. While in our study we simply use this observation as an empirical motivation for addressing the high-frequency nature of tabular dataset target functions, it would be interesting to extend this analysis to more examples and to study the underlying causes of these differences in future work.

## D   Illustration: frequency reduction and the effects of spectral bias

In this appendix, we provide informal analytic arguments, based on the results of [22], illustrating how frequency reduction can improve NN performance given their inherent spectral bias. Theorem 1 of [22] provides the analytic form of the Fourier amplitudes of a general ReLU network, $f(\mathbf{x})$. The authors also show that along each direction of $\mathbf{k}$ space, these amplitudes are upper bounded as $|\widetilde{f}_\theta(\mathbf{k})| \leq N_f L_f(\theta) k^{-\Delta-1}$. Here, $L_f$ is the Lipschitz constant of the NN for a given set of parameters $\theta$, $N_f$ is the number of linear regions, and $1 \leq \Delta \leq d$ depends on the orientation of $\mathbf{k}$ with respect to the polytope faces represented by the NN. In any realistic setting, there is a maximum $N_f L_f$ that can be achieved through training, and therefore the amplitude of the NN Fourier coefficient for a fixed $\mathbf{k}$ is bounded from above. Along each $\mathbf{k}$ direction, assume that the target function behavior at high frequencies falls off more slowly than $k^{-\Delta-1}$. Then, for large enough $k$ the target function Fourier amplitudes, $\widetilde{y}(\mathbf{k})$, cannot be fit by the neural network with fixed architecture and parameters. Let $\Omega$ denote the corresponding high-frequency region of $\mathbb{R}^n$. Reducing the $L^2$ norm of the target function Fourier amplitude over $\Omega$, $\int_\Omega |\widetilde{y}(\mathbf{k})|^2 d\mathbf{k}$, relative to the corresponding integral over $\mathbb{R}^n$ (assuming the target function Fourier amplitudes are square-integrable) will tend to reduce the corresponding error arising from this spectral bias when evaluated on a particular set of data points. Our definition of frequency reduction in terms of the NUDFT amplitudes can be viewed as a proxy for reducing this $L^2$ norm.

Applying the scale transformation with scale factor $a > 1$ can lead to the aforementioned reduction of relative spectral energy over $\Omega$. The Fourier amplitudes for a function $g(\mathbf{x})$ are related to those for the corresponding function, $g_{\text{scaled}}(\mathbf{x})$ acting on scaled inputs as $\widetilde{g}(\mathbf{k}) = 1/a \times \widetilde{g}_{\text{scaled}}(\mathbf{k}/a)$. This relationship shows directly that scale with $a > 1$ maps a given Fourier component of the original function to a component at reduced frequency (and reduced overall magnitude) after applying the scaling transformation. Thus, denoting the corresponding high-frequency domain of the scaled target function as $\Omega_{\text{scaled}}$, in general $\Omega_{\text{scaled}} \subset \Omega$ (again assuming $a > 1$ and that the original target function decays more slowly at large $k$ than the spectral bias bound). The smaller domain of integration, together with the relationship between $\widetilde{y}$ and $\widetilde{y}_{\text{scaled}}$, can reduce the relative amplitude of the spectral integral. The corresponding argument for rank further depends on the underlying distribution of the feature. Consider the simple example of a uniformly-distributed feature $x$ over an interval $[x_1, x_2]$. Then $\text{rank}(x)$ acts in precisely the same way as scale with $a = 1/(x_2 - x_1)$, and the same effect is seen (assuming $x_2 - x_1 < 1$). Similar arguments can be made with other assumptions about the underlying distribution.

These arguments suggest how the transformations we consider can mitigate the impact of NN spectral bias, however they do not account for the other potentially adverse effects of rank and scale on the optimization process. In practice, these effects are important and motivate the learnable convex combination of rank and scale we propose, as it is implicitly regularized by the loss function itself. We plan to analyze the corresponding training dynamics analytically, and further formalize the illustrations above, in future work.

## E   Experiment Details

**Datasets and Training.** We train and evaluate all models across 10 random seeds, and report mean metrics with their 95% confidence intervals. We use 70% of each dataset for training, 15% for validation and 15% for testing. We report evaluation metrics on the test set that correspond to the lowest validation set loss for each model. We use simple *target-encoding* [21] to convert categorical variables to numerics, in order to avoid introducing potential interactions between categorical embedding techniques and our metrics of interest.

**Hyperparameter Spaces.** We conduct hyperparameter tuning for every model across 100 configurations using the validation set. We use HyperOpt [4] to tune the hyperparameters of all approaches

| Parameter | Distribution |
|---|---|
| depth | $\mathrm{UniformInt}[2, 4, 8]$ |
| width | $\mathrm{UniformInt}[128, 256, 512]$ |
| learning rate | $\mathrm{LogUniform}[10^{-6}, 10^{-2}]$ |

Table 3: Hyperparameter space for MLPs. Batch size is set to 128 and is not tuned. PyTorch's [20] implementation of Adam [17] with its default parameters is used for optimization.

| Parameter | Distribution |
|---|---|
| max depth | $\mathrm{UniformInt}[5, 10, 20]$ |
| num. estimators | $\mathrm{UniformInt}[100, 500, 1000]$ |
| learning rate | $\mathrm{LogUniform}[10^{-5}, 10^{-1}]$ |

Table 4: Hyperparameter space for XGBoost. The rest of the XGBoost parameters are set to the defaults of the DMLC XGBoost package [7].

| Parameter | Distribution |
|---|---|
| $n_d = n_a$ | $\mathrm{UniformInt}[56, 60, 64]$ |
| $n_{\mathrm{steps}}$ | $\mathrm{UniformInt}[1, 2, 3]$ |
| learning rate | $\mathrm{LogUniform}[10^{-5}, 10^{-2}]$ |

Table 5: Hyperparameter space for TabNet. Following the suggestion of the authors, we use the same value for $n_d$ and $n_a$, and set the batch size to 1024. PyTorch's [20] implementation of Adam [17] with its default parameters is used for optimization.

considered. Since our main objective is to surface the impact of a frequency-reducing inductive bias and not to perform extensive benchmarking, we focus on tuning the parameters that control the model complexity and learning rates. Specifically, the parameter space considered for each model are shown in Tables 3, 4, and 5.

**Baselines.** As a baseline, we evaluate the impact of common normalization techniques on our metrics of interest and present our results in Figure 2. Specifically, we study MLPs with standardization, robust scaling and quantile normalization [21]. We choose these transformations to compare our methods against for two reasons. First, these are the most common transformations used in the tabular deep learning domain. Second, these transformations are instantiations of rank and scale. Therefore, besides normalization, they impact the frequency spectra. We use `scikit-learn` [21] implementations of these transformations with their default hyperparameters. In our experiments we observe that, among these baselines, there is not a single transformation that consistently performs the best across all datasets. This observation is consistent with the existing tabular deep learning literature, where studies that use different datasets find different transformations working the best. In these studies, the quantile transform, standardization and min-max scaling methods are the most commonly used.

**Frequency Reduction and XGBoost.** In addition to our various MLP experiments, we also study the impact of frequency reduction on XGBoost performance. We first train an MLP until convergence, then use the weights $\theta_g$ to transform input data to a lower-frequency form. We then train an XGBoost model using the transformed data. From Table 6, it can be observed that reduction does not improve XGBoost performance. However, it improves simple MLPs to provide better performance than XGB on 3 datasets, and comparable performance on 2 datasets. The proposed method does not improve upon simple MLPs if the baseline performance is already high (i.e., > 99%), which is expected as these datasets do not feature high-frequency information that cannot be learned by MLPs with unit scale.

### E.1 Weights learned by the proposed approach on 14 datasets

We investigate the weights $\theta_g$ learned by our proposed approach on the 14 datasets. Since these weights get updated jointly with the neural network weights $\theta_f$, this investigation is essential to

| dataset | XGB | XGB* | MLP* |
| --- | --- | --- | --- |
| MagicTelescope | **93.74 ± 0.14** | 93.74 ± 0.14 | 93.67 ± 0.15 |
| MiniBooNE | 98.7 ± 0.02 | 98.7 ± 0.02 | **98.79 ± 0.02** |
| bankMarketing | **93.58 ± 0.12** | 93.58 ± 0.12 | 92.66 ± 0.17 |
| california_housing | **97.22 ± 0.1** | 97.22 ± 0.1 | 96.01 ± 0.1 |
| electricity-normalized | **98.81 ± 0.04** | 98.81 ± 0.04 | 94.51 ± 0.1 |
| house_16H | 95.62 ± 0.15 | 95.62 ± 0.15 | **95.68 ± 0.12** |
| kdd-small | 99.28 ± 0.05 | 99.28 ± 0.05 | 99.17 ± 0.08 |
| Diabetes | 71.02 ± 0.09 | **71.03 ± 0.09** | 68.88 ± 0.13 |
| albert | **76.47 ± 0.06** | 76.47 ± 0.06 | 72.8 ± 0.04 |
| credit-card-clients | 78.12 ± 0.21 | 78.11 ± 0.21 | **78.16 ± 0.23** |
| eye_movements | **91.38 ± 0.25** | 91.38 ± 0.25 | 80.34 ± 0.3 |
| jannis | **88.14 ± 0.13** | 88.08 ± 0.15 | 87.77 ± 0.11 |
| phoneme | **96.74 ± 0.15** | 96.73 ± 0.16 | 94.84 ± 0.12 |
| pol | **99.94 ± 0.01** | **99.94 ± 0.01** | **99.94 ± 0.01** |

Table 6: Un-normalized AUROCs for XGBoost (XGB), XGBoost with low-frequency embeddings (XGB*) and MLP with low-frequency embeddings (MLP*).

understand whether or not $\theta_g$ merely works as a set of additional trainable parameters that adapt to the changes in the rest of the network parameters. If theta $\theta_g$ depends on $\theta_f$, then the low-frequency mappings produced using $\theta_g$ may not be useful as general purpose representations. Figures 8 and 9 depict the weights learned by our proposed approach. Shaded areas represent the 95% confidence intervals across 10 random seeds. Notably, neural network weights $\theta_f$ get initialized differently across the seeds, while $\theta_g$ gets initialized to the same values. Specifically, we initialize all scaling weights as 1, and all ranking weights to 0.5. More advanced initialization techniques may be explored to further improve performance. From the figures, we observe that $\theta_g$ converges to similar values across different random seeds on all the datasets considered. Additionally, we observe that for the datasets *bank-marketing (bank)* and *diabetes*, some of the scaling weights converge to zero, eliminating the use of the corresponding features and causing the combining weights to not get updates (i.e., staying at 0.5). This behavior may be interpreted as a frequency-reducing feature selection mechanism: if a feature's contribution to the performance is not sufficiently high, compared to its contribution to the frequency of the decision boundary, the neural network may choose to drop it. This behavior should be further investigated and compared to methods that were designed to select informative features. Overall, the results in Figures 8 and 9, along with Figure 2, suggest that our proposed method is indeed enabling the network to learn useful representations of the data biased towards lower-frequency target functions.

### E.2 Un-normalized AUROCs on all dataset and approaches considered

We provide un-normalized AUROCs of all approaches considered in Table 7. The methods unit, standard, robust, quantile and selective-quantile are applied to MLPs. Based on our experimental results, we observe that the best performing baseline varies across different datasets. As a result, in order to produce a comparison of the overall performance, we provide average normalized metrics in Figure 2.

## F  Computational Complexity

Our proposed method introduces the additional parameters $\theta_g : \{\theta_g^s, \theta_g^r, \theta_g^c\}$. The scaling and combining weights, $\theta_g^s$ and $\theta_g^c$ are learned together with the neural network weights $\theta_f$. The parameters $\theta_g^r$ are set by fitting the ranking transformer to the training set as a preprocessing step (i.e., only once), and extracting a mapping from reference points in the training sets to their ranks. We sample 1000 points to extract this mapping.

In all experiments, we allocate two scaling weights (for raw and ranked features), and one combining weight for each feature. Therefore, the additional parameter complexity introduced to a neural network with input size $N$ is $\mathcal{O}(N)$. In comparison, an additional fully connected layer introduces

| dataset | XGB | XGB* | TabNet | TabNet* | MLP* |
|---|---|---|---|---|---|
| MagicTelescope | 93.74 ± 0.14 | 93.74 ± 0.14 | 90.08 ± 1.54 | 91.56 ± 0.59 | 93.67 ± 0.15 |
| MiniBooNE | 98.7 ± 0.02 | 98.7 ± 0.02 | 89.47 ± 0.65 | 98.59 ± 0.03 | 98.79 ± 0.02 |
| bankMarketing | 93.58 ± 0.12 | 93.58 ± 0.12 | 91.74 ± 0.19 | 89.22 ± 0.51 | 92.66 ± 0.17 |
| california_housing | 97.22 ± 0.1 | 97.22 ± 0.1 | NA | NA | 96.01 ± 0.1 |
| electricity-normalized | 98.81 ± 0.04 | 98.81 ± 0.04 | 84.45 ± 2.58 | 84.72 ± 1.97 | 94.51 ± 0.1 |
| house_16H | 95.62 ± 0.15 | 95.62 ± 0.15 | 89.24 ± 1.17 | 94.54 ± 0.13 | 95.68 ± 0.12 |
| kdd-small | 99.28 ± 0.05 | 99.28 ± 0.05 | 96.62 ± 0.73 | 98.72 ± 0.15 | 99.17 ± 0.08 |
| Diabetes | 71.02 ± 0.09 | 71.03 ± 0.09 | NA | NA | 68.88 ± 0.13 |
| albert | 76.47 ± 0.06 | 76.47 ± 0.06 | NA | NA | 72.8 ± 0.04 |
| credit-card-clients | 78.12 ± 0.21 | 78.11 ± 0.21 | NA | NA | 78.16 ± 0.23 |
| eye_movements | 91.38 ± 0.25 | 91.38 ± 0.25 | NA | NA | 80.34 ± 0.3 |
| jannis | 88.14 ± 0.13 | 88.08 ± 0.15 | NA | NA | 87.77 ± 0.11 |
| phoneme | 96.74 ± 0.15 | 96.73 ± 0.16 | 89.14 ± 0.28 | 81.28 ± 2.9 | 94.84 ± 0.12 |
| pol | 99.94 ± 0.01 | 99.94 ± 0.01 | 77.18 ± 6.46 | 94.41 ± 2.45 | 99.94 ± 0.01 |

| dataset | unit | standard | robust | quantile | selective-quantile |
|---|---|---|---|---|---|
| MagicTelescope | 93.32 ± 0.16 | 93.27 ± 0.18 | 93.52 ± 0.17 | 93.27 ± 0.18 | 93.37 ± 0.18 |
| MiniBooNE | 96.47 ± 0.02 | 98.29 ± 0.02 | 98.77 ± 0.02 | 98.74 ± 0.02 | 98.72 ± 0.02 |
| bankMarketing | 92.71 ± 0.14 | 92.26 ± 0.15 | 91.82 ± 0.16 | 92.11 ± 0.15 | 92.4 ± 0.14 |
| california_housing | 95.56 ± 0.1 | 95.75 ± 0.09 | 95.66 ± 0.09 | 95.65 ± 0.12 | 95.63 ± 0.09 |
| electricity-normalized | 93.04 ± 0.12 | 92.99 ± 0.12 | 92.65 ± 0.16 | 93.85 ± 0.12 | 94.64 ± 0.11 |
| house_16H | 94.68 ± 0.13 | 95.23 ± 0.13 | 95.53 ± 0.14 | 95.25 ± 0.14 | 95.5 ± 0.16 |
| kdd-small | 99.32 ± 0.05 | 99.16 ± 0.06 | 99.21 ± 0.07 | 99.28 ± 0.05 | 99.34 ± 0.05 |
| Diabetes | 69.65 ± 0.09 | 69.29 ± 0.09 | 69.57 ± 0.11 | 67.96 ± 0.11 | 69.54 ± 0.09 |
| albert | 72.16 ± 0.06 | 72.39 ± 0.06 | 71.99 ± 0.04 | 72.27 ± 0.04 | 72.27 ± 0.05 |
| credit-card-clients | 77.14 ± 0.25 | 77.67 ± 0.24 | 77.73 ± 0.2 | 77.39 ± 0.21 | 77.56 ± 0.27 |
| eye_movements | 79.86 ± 0.29 | 80.62 ± 0.34 | 79.53 ± 0.28 | 75.77 ± 0.3 | 76.98 ± 0.29 |
| jannis | 86.67 ± 0.11 | 87.32 ± 0.11 | 87.39 ± 0.11 | 86.84 ± 0.12 | 86.99 ± 0.13 |
| phoneme | 94.13 ± 0.25 | 94.71 ± 0.13 | 94.18 ± 0.23 | 94.11 ± 0.26 | 94.12 ± 0.22 |
| pol | 99.96 ± 0.0 | 99.96 ± 0.0 | 99.94 ± 0.01 | 99.32 ± 0.03 | 99.94 ± 0.01 |

Table 7: Un-normalized AUROCs of all approaches considered. An asterisk (*) indicates that the corresponding model uses the proposed low-frequency transformations. Note that our TabNet experiments only consider binary classification datasets.

$\mathcal{O}(d_h N)$, where $d_h$ is the width of the subsequent layer. Notably, our approach can be extended to a high-dimensional setting similar to [11] to map each feature to a $k$-dimensional vector that scales and combines the same feature with its ranked counterpart in different ratios. Although this extension is out of the scope of this work, the complexity of this approach would be $\mathcal{O}(kN)$, where our current approach sets $k := 1$.

## G  Limitations

In this paper, we focus on building the connection between the spectral bias of neural networks and its impact on tabular deep learning performance. We show that heterogeneous tabular datasets tend to feature high-frequency target functions as defined by the Fourier analysis methods presented. We also observe that with frequency-reducing transformations, these target functions can be smoothed to alleviate the negative impact of spectral bias. Consequently, we propose using frequency reduction as inductive bias while learning with neural networks. We discuss some of the limitations of our methods and analysis below.

**Limitations of the proposed method.** We implement the proposed inductive bias as a neural network layer that learns to scale and map points along a linear trajectory between (scaled) input features and their ranked counterparts. This implementation is motivated by our empirical analyses of commonly-used transformations in tabular deep learning, and is an attempt to generalize them while retaining their frequency-reducing properties. However, our implementation only covers a subset of such transformations. Indeed, one may carefully design transformations that guarantee frequency reduction with fewer negative side effects on the other training dynamics. Also, we limit

our experiments to independently transform each feature. More beneficial transformations might be found by instead first grouping features with informative interactions (i.e., subsets that are compatible to form informative higher-order features), then attempting to transform them to a low-frequency form. As tabular datasets feature heterogeneous feature spaces, identification of such compatible feature sets may be a challenge. It is possible that, for example, a pair of features may meaningfully interact in certain subregions (i.e., subdomains). Then, one may choose to identify these regions and represent them in different dimensions of the feature space. Such a mapping of features onto high-dimensional spaces to may help reduce target function frequency by transforming the feature space to be more homogeneous. We do not consider such approaches in this work. Finally, our methods only transform the inputs to the network, while more generally one may apply our frequency reduction inductive bias to all network activations to ensure that learned representations also have low-frequency form.

**Limitations of the experiments.** Our experiments mainly focus on observing the dataset-dependent effects of existing transformations that reduce target function frequency, and demonstrating how this dependency can be alleviated by using frequency reduction as an inductive bias during learning. In doing so, we limited our analysis to classification problems, although our methods can be applied in regression settings as well. Also, although our experimental setup enables a fair performance comparison between MLPs with different transformations, the corresponding results may not reflect the best performance achievable by MLPs. This is because we do not use techniques such as batch normalization, residual connections and weight decay regularizers that are shown to improve MLP performance on tabular data. This scope will be extended in future work. Our experiments with other neural network architectures are also limited to TabNet on binary classification datasets. Future work will also include extensive experiments with additional tabular data-specific deep learning architectures in order to explore the relationship between the improvements they offer and the frequency spectra of the functions they learn. Finally, although we qualitatively observe that our proposed methods alleviate the negative effects of spectral bias through our analysis of Fourier spectra and the corresponding impact on performance for all datasets considered, our quantification of spectral bias (see Appendix B.1) is limited to synthetic datasets due to its computational complexity. Such quantification can be extended beyond synthetic data experiments in the future.

# H  Implementation

In this section, we demonstrate how to implement the key components of our analyses and proposed method using PyTorch [20].

## H.1  Proposed method as a network layer

The module `CustomLayer` implements our frequency-reducing transformation. In our implementation, in addition to scaling and combining, we also incorporate `shifting` factors in order to center features when necessary. We empirically observed that addition of the shifting factors did not change the performance of the neural network significantly. Notably, this implementation assumes that the input is in the form $\mathbf{x} = [\mathbf{x}_{\text{src}}, \mathbf{x}_{\text{trg}}] = [\mathbf{x}_{\text{raw}}, \mathbf{x}_{\text{ranked}}]$, where ranked is generated using `scikit-learn`'s quantile transform. Also, for the "raw" features, our implementation uses their scaled version as an input to support training stability.

```python
class CustomLayer(nn.Module): # proposed layer
    def __init__(self, size):

        self.size = size
        weights = torch.Tensor(torch.ones((1, size)) * 0.5)
        scaling_factors_src = torch.Tensor(torch.ones((1, size)))
        shifting_factors_src = torch.Tensor(torch.zeros((1, size)))
        scaling_factors_trg = torch.Tensor(torch.ones((1, size)))
        shifting_factors_trg = torch.Tensor(torch.zeros((1, size)))

        self.weights = nn.Parameter(weights)
        self.scaling_weights_src = nn.Parameter(scaling_factors_src)
        self.shifting_weights_src = nn.Parameter(shifting_factors_src)
        self.scaling_weights_trg = nn.Parameter(scaling_factors_trg)
        self.shifting_weights_trg = nn.Parameter(shifting_factors_trg)
```

```python
    def forward(self, x):
        x_src = x[:, :x.size()[1]//2] # raw input
        x_trg = x[:, x.size()[1]//2:] # ranked input

        w = self.weights
        c_src = self.scaling_weights_src
        u_src = self.shifting_weights_src
        c_trg = self.scaling_weights_trg
        u_trg = self.shifting_weights_trg

        x = (((x_src + u_src) * c_src) * w) + (((x_trg + u_trg) * c_trg) * (1 - w))
        return x

class MLP(nn.Module):
    def __init__(self, layers, activation_layer, p_dropout, bias, num_classes):

        layer_list = [nn.Linear(layers[i], layers[i+1], bias=bias) for i in
            range(len(layers)-1)]
        self.layers = nn.ModuleList(layer_list)
        self.classifier = nn.Linear(layers[-1], num_classes)
        self.dropout = nn.Dropout(p=p_dropout)
        self.activation_layer = activation_layer
        self.customLayer = CustomLayer(layers[0])

    def forward(self, x):
        x = self.customLayer(x)

        for i in range(len(self.layers)):
            x = self.layers[i](x)
            x = self.activation_layer(x)
            x = self.dropout(x)

        yhat = self.classifier(x)

        return yhat
```

## H.2    Functions to calculate spectra

To efficiently compute NUDFT amplitudes, we utilize the `nfft_adjoint` function from the third party package `nfft` (https://github.com/jakevdp/nfft). The implementation is as follows:

```python
# !pip install nfft
from nfft import nfft_adjoint

def NUDFT(x, y, kmax, nk):
    return (1/len(x)) * nfft_adjoint(-(x*kmax/nk), y, 2*(nk+1))[nk+1:]

def spectrum_NUDFT(x, y, kmax=50, nk=1000):
    kvals = np.linspace(0, kmax, nk+1)
    nufft = NUDFT(x, y, kmax, nk)
    return [kvals, np.array(nufft, dtype="complex_")]
```

## H.3    Selective-rank

In our experiments with selective rank, we set `transformer` to `scikit-learn`'s quantile transform.

```python
def get_features_with_reduced_frequency(X_tr, y_tr, transformer, threshold=0.75):

    features = []
    no_transform_features = []
```

```python
        X_tr_quantile = transformer.fit_transform(X_tr)

        for i in range(X_tr.shape[1]):
            v_raw = spectrum(X_tr[:, i], y_tr)[lower_bound:]
            v_qnt = spectrum(X_tr_quantile[:, i], y_tr)[lower_bound:]
            metric = np.mean(abs(v_qnt) ** 2)/np.mean(abs(v_raw) ** 2)

            if metric <= threshold:
                features.append(i)
            else:
                no_transform_features.append(i)

    return features

def selective_transform(dataset, scaler, config, transformer):
    X_tr, y_tr = dataset['tr']['X'], dataset['tr']['y']
    X_va, y_va = dataset['va']['X'], dataset['va']['y']
    X_te, y_te = dataset['te']['X'], dataset['te']['y']

    transform_cols = get_features_with_reduced_frequency(X_tr, y_tr, transformer,
        threshold=config['f_threshold'])

    ct = ColumnTransformer([("sel-rank", transformer, transform_cols)])
    ct.fit(X_tr, y_tr)

    X_tr[:, transform_cols] = ct.transform(X_tr)
    X_va[:, transform_cols] = ct.transform(X_va)
    X_te[:, transform_cols] = ct.transform(X_te)

    return X_tr, X_va, X_te
```

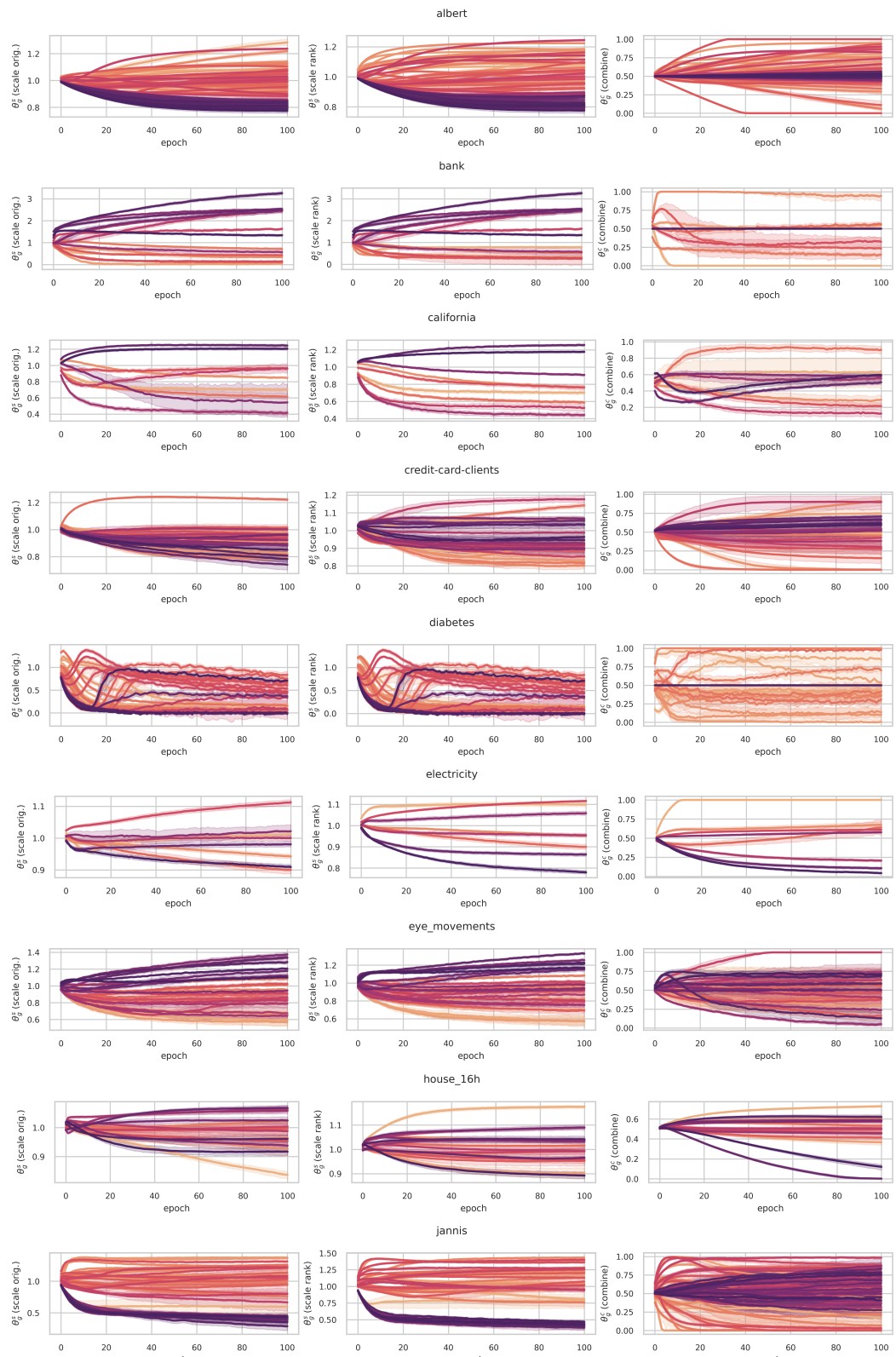

Figure 8: Weights learned by the proposed approach on all 14 datasets, part 1.

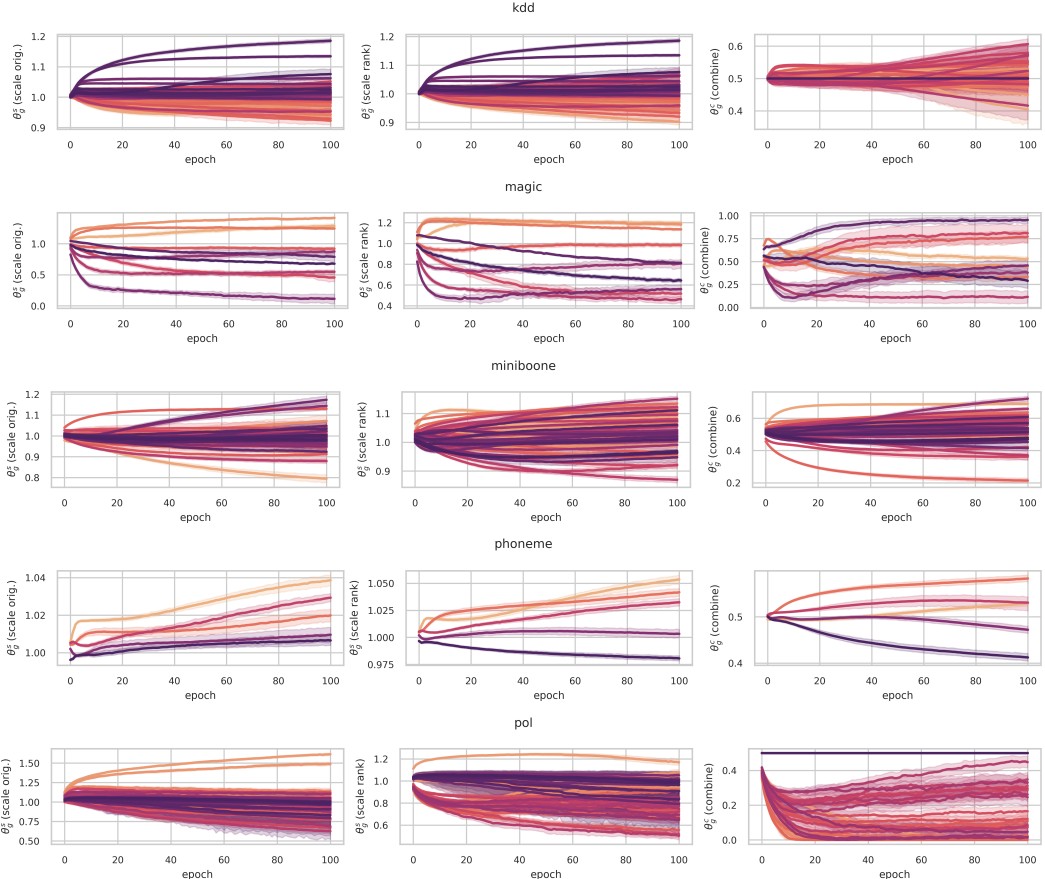

Figure 9: Weights learned by the proposed approach on all 14 datasets, part 2.

