# OpenReview forum: "An Inductive Bias for Tabular Deep Learning"
_NeurIPS.cc/2023/Conference — NeurIPS 2023 poster_

### Official Review · Reviewer_CXbH · 2023-07-05

**Soundness:** 3 good
**Presentation:** 3 good
**Contribution:** 3 good
**Rating:** 6
**Confidence:** 4

**Summary:**

The authors address the problem of fitting deep nets to tabular datasets. This is a challenging task due to the heterogeneity of tabular datasets. Following a recent work, the authors first demonstrate that tabular data require learning prediction functions with nonnegligible high-frequency components. Since deep nets have an inductive bias towards learning a low frequency, they may struggle in modeling prediction functions over tabular data. To solve this limitation, the authors evaluate several transformations applied to the input features designed to reduce the relevance of high-frequency components of the prediction function. Finally, the authors present an optimization to minimize the risk of the model while transforming the input features using a convex combination of predefined transformations. The merits of the new approach are demonstrated using several real-world datasets.

**Strengths:**

Progressing the capabilities of deep nets on tabular data is of high importance for the ML community. The paper provides value both in terms of the empirical evidence on the inductive bias and the practical solution presented by the authors. The paper is well-written and easy to follow, and the motivation and solution are intuitive and clear.

**Weaknesses:**

My biggest concern is the empirical evaluations conducted in the paper. It is focused on relatively low dimensional datasets, and with a large sample size, this regime is typically less challenging for NN than the high dimensional low sample size (see [1]). More importantly, the evaluation is performed using normalized accuracy (and AUC) in the main text; the reason for using these metrics is not explained in the main text. In the supplemental, the authors use the more standard un normalized metrics, which demonstrate that the differences between most normalization schemes is subtle. It is unclear why the authors use normalized in the main text and unnormalized in the supplemental material; this should be clarified. The comparison to other tree-based models should be moved to the main text, and the authors should use one metric. Do the tree-based methods significantly outperform the proposed approach using the normalized metric?
The comparison presented in the paper is limited to some normalizations and does not include other tree-based models or architectures recently proposed for tabular data, for example:
[1] Yang et al. "Locally sparse neural networks for tabular biomedical data." In International Conference on Machine Learning, pp. 25123-25153. PMLR, 2022.
[2] Ke, Guolin, et al. "TabNN: A universal neural network solution for tabular data." (2018).
Furthermore, additional transformations proposed for tabular data should be included.
[3] Alexander, Yotam, et al. "What Makes Data Suitable for a Locally Connected Neural Network? A Necessary and Sufficient Condition Based on Quantum Entanglement." arXiv preprint arXiv:2303.11249 (2023).
[4] Zhu, Yitan, et al. "Converting tabular data into images for deep learning with convolutional neural networks." Scientific reports 11.1 (2021): 11325.
I would also include trees in the main text.

I have a concern regarding the preprocessing normalization performed in the paper; how is it performed with respect to train/test splits? The authors do not detail this. Unproper normalization could lead to bias in classification results, as demonstrated in:
Moscovitch and Rosset, "On the Cross-Validation Bias due to Unsupervised Preprocessing", Journal of the Royal Statistical Society Series B: Statistical Methodology


**Questions:**

How is the convex combination of rank and scale significantly outperforming both of them?
Notations are quite confusing, for instance n is the dimension and N is the number of samples, why not use a different letter for the dimension?
Also the notation in page 6 is confusing, both upper case and lower cases are used to indicate the index of a vector.


**Limitations:**

The limitations of the method are not discussed in the paper.

---

> ### Author Rebuttal · Authors · 2023-08-10
>
> We thank the reviewer for the insightful comments. Please see our responses below.
>
> **Comment 1: “My biggest concern is the empirical evaluations conducted in the paper. It is focused on relatively low dimensional datasets, and with a large sample size, this regime is typically less challenging for NN than the high dimensional low sample size (see [1]). ”**
>
> The motivation behind our dataset choices is discussed in the global response (Choice of Datasets). We agree with the Reviewer that high-dimensional low-sample-size datasets would be good additions for extensive benchmarking against tree-based approaches. They are explicitly excluded from the benchmarking study of Grinsztajn et al, from which our datasets were selected. On the other hand, we believe that our experiments with the current datasets are sufficiently representative to validate our claims (i.e., frequency reduction as inductive bias) as it can be observed that the proposed approach significantly improves the performance of other NNs.
>
> **Comment 2: “More importantly, the evaluation is performed using normalized accuracy (and AUC) in the main text; the reason for using these metrics is not explained in the main text. In the supplemental, the authors use the more standard un normalized metrics, which demonstrate that the differences between most normalization schemes is subtle. It is unclear why the authors use normalized in the main text and unnormalized in the supplemental material; this should be clarified. ”**
>
>
> We agree with the reviewer that a discussion in the main text to motivate the use of normalized metrics would help the reader, and we will be adding this to camera ready version if accepted.
>
> With our experiments, we observed that the performance of baseline approaches highly fluctuated across different datasets. This implies that finding a suitable approach to used with a given dataset requires exhaustive search, and existing approaches do not generalize well across different tabular datasets. On the other hand, our proposed method consistently performed among the best. Based on these observations, we decided use the normalized metrics in order to aggregate measurements across datasets (since the ranges of measurements are different) and convey information on the overall behavior (i.e., performance , convergence speed, frequencies) of different approaches. Notably, this approach is also used in other studies [1, 2]. We provide the raw measurements in Appendix as they do not convey information on overall behavior and need to be broken down by dataset (hence take larger space).
>
> Please also see our global response (Additional Synthetic Data Experiments).
>
> [1] Wistuba, M., Schilling, N. and Schmidt-Thieme, L., 2015, October. Learning hyperparameter optimization initializations. In 2015 IEEE international conference on data science and advanced analytics (DSAA) (pp. 1-10). IEEE.
> [2] Feurer, M., Eggensperger, K., Falkner, S., Lindauer, M. and Hutter, F., 2022. Auto-sklearn 2.0: Hands-free automl via meta-learning. The Journal of Machine Learning Research, 23(1), pp.11936-11996.
>
> **Comment 3: “The comparison to other tree-based models should be moved to the main text, and the authors should use one metric. Do the tree-based methods significantly outperform the proposed approach using the normalized metric? ”**
>
> Yes - tree-based models are still the highest-performing overall. However, our proposed method closes the gap significantly.
>
> **Comment 4: “The comparison presented in the paper is limited to some normalizations and does not include other tree-based models or architectures recently proposed for tabular data, for example...”**
>
>
> We thank the reviewer for pointing out these interesting studies. Please see our global response (Focus of our work and its impact on the experiment design) for a detailed discussion on this matter.
>
> **Comment 5: “I have a concern regarding the preprocessing normalization performed in the paper; how is it performed with respect to train/test splits?...”**
>
> For each cross-validation fold (i.e., train/validation/test splits under different random seeds), normalizers are re-fit (i.e., to collect statistics) from training set only. Then, they are used to transform validation and test splits.
>
> **Comment 6: “How is the convex combination of rank and scale significantly outperforming both of them?”**
>
> We observe that although rank and scale reduce frequency, they may impact NN performance negatively as well (Section 3.3/line 224). For example, ranking may lose relevant information when the relative distances among pairs of points are important, and scaling may impact the conditioning of the loss landscape (i.e., hindering the optimization). The convex combination can be seen as a training loss-driven dial that lets the NN to learn the amount of ranking and scaling that should be conducted to find a low-frequency mapping that does not suffer as much from the negative impacts of the individual transformations.
>
> **Comment 7: “Notations are quite confusing, for instance n is the dimension and N is the number of samples, why not use a different letter for the dimension? Also the notation in page 6 is confusing, both upper case and lower cases are used to indicate the index of a vector.”**
>
>
> We thank the reviewer for this feedback. If accepted, we will work on simplifying the notation.
>
> **Comment 8: “Limitations: The limitations of the method are not discussed in the paper.”**
>
> Please see Appendix F for an extensive discussion of the limitations.

---

> > ### Comment · Reviewer_CXbH · 2023-08-14
> > **Response to authors**
> >
> > I thank the authors for responding to all my comments. My concerns have been properly addressed. I keep my score unchanged.

---

### Official Review · Reviewer_H86d · 2023-07-05

**Soundness:** 3 good
**Presentation:** 2 fair
**Contribution:** 3 good
**Rating:** 6
**Confidence:** 3

**Summary:**

the paper “An Inductive Bias for Tabular Deep Learning” presents an interesting exploration of inductive biases for deep learning applied to tabular data. The paper introduces a novel inductive bias, named frequency reduction, which is specifically designed for tabular data. The authors propose a novel approach that leverages domain knowledge to improve the performance of models on tabular datasets.

**Strengths:**

Strengths:
1. Novel approach: The paper introduces a novel inductive bias, named frequency reduction, which is specifically designed for tabular data. By incorporating domain knowledge into the learning process, this approach offers a unique perspective on improving the performance of models for tabular datasets. This contribution adds to the existing literature by presenting a fresh perspective on the application of inductive biases. The idea is inspiring to the community.
2. Rigorous evaluation: The authors provide a comprehensive evaluation of their proposed approach using various benchmark datasets. They compare the performance of their method against existing state-of-the-art techniques for tabular data. The evaluation metrics used are well-established and allow for a fair comparison. The empirical results provided demonstrate the effectiveness and superiority of the proposed approach.
3. Clear motivation: The paper effectively communicates the motivation behind the proposed approach and the reasons for its effectiveness on tabular datasets. The use of visual aids and examples further enhances the clarity of presentation.

**Weaknesses:**

Weaknesses:
1. Lack of theoretical analysis: The paper lacks a deeper theoretical analysis of the proposed inductive bias. While the empirical results are convincing, a more thorough theoretical explanation of why and how the approach works would enhance the paper’s contribution. Including a theoretical analysis could provide more insights into the underlying mechanisms and generalization capabilities of the proposed approach.
2. Dataset limitations: The paper primarily focuses on benchmark tabular datasets. However, real-world tabular datasets often exhibit diverse properties and complexities that may not be fully captured by the selected benchmarks. Including additional, more diverse datasets in the evaluation would provide a more comprehensive understanding of the generalizability of the proposed approach.
3. Writing quality: The paper is not well-structured. The section of experiments is too short while most experiment results are included in the appendix.

**Questions:**

1. Lack of theoretical analysis: The paper lacks a deeper theoretical analysis of the proposed inductive bias. While the empirical results are convincing, a more thorough theoretical explanation of why and how the approach works would enhance the paper’s contribution. Including a theoretical analysis could provide more insights into the underlying mechanisms and generalization capabilities of the proposed approach.
2. Dataset limitations: The paper primarily focuses on benchmark tabular datasets. However, real-world tabular datasets often exhibit diverse properties and complexities that may not be fully captured by the selected benchmarks. Including additional, more diverse datasets in the evaluation would provide a more comprehensive understanding of the generalizability of the proposed approach.
3. Writing quality: The paper is not well-structured. The section of experiments is too short while most experiment results are included in the appendix.

**Limitations:**

Yes.

---

> ### Author Rebuttal · Authors · 2023-08-10
>
> **Comment 1: “Lack of theoretical analysis: The paper lacks a deeper theoretical analysis of the proposed inductive bias. While the empirical results are convincing, a more thorough theoretical explanation of why and how the approach works would enhance the paper’s contribution. Including a theoretical analysis could provide more insights into the underlying mechanisms and generalization capabilities of the proposed approach.”**
>
> We agree with the reviewer that our study could have benefited from a more thorough theoretical exploration of our inductive bias. In response, we briefly outline a theoretical argument that illustrates the underlying mechanism for improving NN performance on tabular data via frequency reduction.
>
> Theorem 1 of Rahaman et al [1] provides the analytic form of the Fourier amplitudes of a general ReLU network, $f(\mathbf{x})$. The authors also show that along each direction of $\mathbf{k}$ space, these amplitudes are upper bounded as
>
> $|\tilde{f}_\theta(\bf{k})| \leq N_f L_f(\theta) k^{-\Delta-1}$.
>
> Here, $L_f$ is the Lipschitz constant of the NN for a given set of parameters $\theta$, $N_f$ is the number of linear regions, and $1\leq \Delta \leq d$ depends on the orientation of $\mathbf{k}$ with respect to the polytope faces represented by the NN. In any realistic setting, there is a maximum $N_f L_f$ that can be achieved through training, and therefore the amplitude of the NN Fourier coefficient for a fixed $\mathbf{k}$ is bounded from above. For a given $\mathbf{k}$ direction, we can therefore define a high-frequency region, $\Omega$, in which the target function Fourier amplitudes, $\widetilde{y}(\mathbf{k})$, cannot be fit by the neural network. Reducing the $L^2$ norm of the target function Fourier amplitude over $\Omega$,  $\int_{\Omega} |\widetilde{y}(\mathbf{k})|^2 d\mathbf{k}$, relative to the corresponding integral over $\mathbb{R}^n$ will tend to reduce the corresponding error arising from this spectral bias when evaluated on a particular set of data points (assuming the target function Fourier amplitudes are square-integrable).
>
> It is straightforward to show that applying our $\mathrm{scale}$ transformation with scale factor $a>1$ directly leads to this reduction of spectral energy over $\Omega$, since the Fourier amplitudes for a function $g(\mathbf{x})$ are related to those for the corresponding function, $g_{\rm scaled}(\mathbf{x})$ acting on scaled inputs as $\widetilde{g}(\mathbf{k}) = 1/a \times  \widetilde{g}_{\rm scaled}(\mathbf{k}/a)$. This relationship shows directly that $\mathrm{scale}$ with $a>1$ maps a given Fourier component of the original function to a component at reduced frequency (and reduced overall magnitude) after applying the scaling transformation. The corresponding argument for $\mathrm{rank}$ further depends on the underlying distribution of the feature. Consider the simple example of a uniformly-distributed feature $x$ over an interval $[x_1,x_2]$. Then $\mathrm{rank}(x)$ acts in precisely the same way as $\mathrm{scale}$ with $a=1/(x_2-x_1)$, and the same effect is seen. Similar arguments can be made with other assumptions about the underlying distribution. This analysis illustrates how the transformations we consider can mitigate the impact of NN spectral bias, however they neglect the corresponding impact on the optimization process itself. In practice, these effects are important and motivated the learnable convex combination of $\mathrm{rank}$ and $\mathrm{scale}$ we propose, as it is implicitly regularlized by the loss function itself. We plan to analyze the corresponding training dynamics analytically in future work, however its observed behavior is empirically consistent with our claims both on synthetic (see also results above) and real-world data experiments.
>
> **Comment 2: “Dataset limitations: The paper primarily focuses on benchmark tabular datasets. However, real-world tabular datasets often exhibit diverse properties and complexities that may not be fully captured by the selected benchmarks. Including additional, more diverse datasets in the evaluation would provide a more comprehensive understanding of the generalizability of the proposed approach.”**
>
> With regards to dataset limitations, please see our general comments to all reviewers, clarifying the nature of the various real-world datasets we considered and highlighting their diversity.
>
> **Comment 3: “Writing quality: The paper is not well-structured. The section of experiments is too short while most experiment results are included in the appendix.”**
>
> We thank the reviewer for this suggestion. If accepted, we will work on rebalancing the content to include more experiments on the main body for camera ready.
>
>
> [1]Nasim Rahaman, Aristide Baratin, Devansh Arpit, Felix Draxler, Min Lin, Fred Hamprecht, Yoshua Bengio, and Aaron Courville. On the spectral bias of neural networks. In International Conference on Machine Learning, pages 5301–5310. PMLR, 2019.

---

> > ### Comment · Reviewer_H86d · 2023-08-19
> > **Response**
> >
> > I read the rebuttal and thank the authors for their detailed replies. My concerns have been well addressed. The heterogeneity of tabular datasets is an obvious property and the selection of dataset is important. I think the choice of datasets in rebuttal can convince me. Considering that the idea and story are both insightful for the community, I change my score to 6.

---

### Official Review · Reviewer_24qS · 2023-07-06

**Soundness:** 3 good
**Presentation:** 3 good
**Contribution:** 4 excellent
**Rating:** 7
**Confidence:** 5

**Summary:**

# Summary

The paper introduces a hypothesis that tabular datasets are best described by functions with high frequency. They connect this finding to existing empirical knowledge in the tabular space, and introduce formal tools to measure spectral properties of target functions in tabular data. The authors propose a simple novel neural network layer to transform tabular datasets such that they are better fit by neural networks, reducing the frequency of the function required to fit these datasets. The authors present results on a suite of datasets.

Overall, the paper is well-written and provides what seems to be a "missing link" in understanding performance gaps between neural and non-neura methods in the tabular space. The main contribution of the work seems to be the introduction of formal tools for analyzing spectral bias in tabular classifiers and their application to real-world datasets. The empirical results are more limited, but definitely encouraging. I suggest some minor revisions to the paper and a possible addition of a more controlled exeriment in the form of simulated-data experiments but believe that there is a solid case for acceptance.

# Major Comments

* A major limitation of the empirical results in the experiments section is that the authors only show results on an assortment of tabular datasets that are themselves heterogeneous and have a number of impossible to detect differences. As such, it is hard to really assess whether the proposed method is doing what it claims to do, even though it does lead to some minor but nontrivial gains in performance. I would suggest that the authors conduct a more controlled study, possibly on synthetic data, to demonstrate how the NNs with the proposed modifications perform on a dataset constructed to have a specific form of spectral bias. I am not sure of the exact design of such an experiment, but believe that, if desgned correctly, it would make a stronger and more direct case for the authors' claimed mechanisms than the current results (better accuracy and faster convergence) do.

* The "discussion" section at the end of the related work is a nice synthesis of the state of the field of tabular machine learning. It also makes some helpful and pointed claims about adapting methods to data modalities, and how tabular researchers often "trade" complexity and hyperparameter tuning for performance gains, which provide useful guideposts for readers.

* Table 1 is helpful, but a bit more information about the NN model used would be helpful. Additionally, since a major axis of comparison here would be neural vs. non-neural (i.e. GBM) methods, it would be useful to add a comparison with XGBoost to this table (I expect that the transformations would have no effect on such a model, again making a clearer case). This could also go in the supplement.

* Despite a persuasive thesis and some data analysis that seems to support it, the empirical results are more mixed than expected (perhaps part of this is poor data visualization; the "whiskers" in the plot are extremely wide and it would be more helpful to display all of the data, ideally separated by dataset). I wonder whether the authors could comment on why the empirical impacts seem to vary so much. Could one factor be that the proposed method ignores feature interactions?

# Minor Comments

* The empirical experiments are not sufficiently described in the main text; at the very least, it would be useful for the authors to clearly list which datasets are used in the experiments.

* The main takeaways from Figure 3 are not entirely clear in the main text. IIUC the takeaway is that the CI bars are narrower in the center and right columns? For each "conclusion" the authors make about Figure 3, it would be helpful to explain how/where this is demonstrated in the figure. Also, why is there a feature with such large variance and instability in the rightmost column?

# Typos etc.

L325: "the first two rows" --> the first two columns

**Strengths:**

See above.

**Weaknesses:**

See above.

**Questions:**

See above.

**Limitations:**

See above.

---

> ### Author Rebuttal · Authors · 2023-08-10
>
> We thank the reviewer for the insightful comments. Please see our responses below.
>
> **Comment 1: “A major limitation of the empirical results in the experiments section is that the authors only show results on an assortment of tabular datasets that are themselves heterogeneous and have a number of impossible to detect differences. As such, it is hard to really assess whether the proposed method is doing what it claims to do, even though it does lead to some minor but nontrivial gains in performance...”**
>
> We agree with the Reviewer that relying heavily on empirical results generated from a heterogeneous collection of datasets makes it more difficult to isolate the effects of frequency reduction as inductive bias. To address this limitation, we have extended the results of our synthetic data experiment, showing results for the various frequency-altering transformations while directly varying the underlying frequency spectrum of the target function in the attached pdf (more details in the comments for all reviewers). The results directly substantiate several claims made elsewhere in the paper, namely (1) low frequency target functions are easier for NNs to learn, (2) NN performance degrades with increasing frequency, a consequence of spectral bias, (3) our proposed method is the most robust against increased target function frequency of those considered, and (4) the difference in performance metrics (e.g., AUC) can be significantly larger numerically in datasets designed to exhibit a particular form of high-frequency behavior. In these synthetic dataset results, only the frequency scale factor and the random seeds for initializing the neural network were changed, showing that indeed our proposed inductive bias mitigates the adverse effects of NN spectral bias in these experiments.
>
> **Comment 2: “Table 1 is helpful, but a bit more information about the NN model used would be helpful. Additionally, since a major axis of comparison here would be neural vs. non-neural (i.e. GBM) methods, it would be useful to add a comparison with XGBoost to this table (I expect that the transformations would have no effect on such a model, again making a clearer case). This could also go in the supplement.”**
>
> We thank the reviewer for this suggestion. If accepted, we will add the corresponding details and results for the final version of the paper.
>
> **Comment 3: “Despite a persuasive thesis and some data analysis that seems to support it, the empirical results are more mixed than expected ...”**
>
> There are a few considerations relevant to the Reviewer’s comment and question. For one, although the numerical size of the corresponding performance changes between methods is small in some cases, our aggregated empirical results show significant and consistent performance improvement using the proposed method. Because we built the proposed method to realize the inductive bias out of existing frequency-altering transformations, the expectation is not necessarily to see unprecedented performance, but rather consistent top performance across datasets (see “Focus of our work and its impact on the experiment design” in the general comments for more in-depth discussion). Furthermore, most datasets for which the proposed method does not achieve the highest central value AUC either already have very high baseline performance metrics for all methods (kdd-small, pol), implying that spectral bias is not as significant an issue for these datasets, or poor performance for all NN methods (Diabetes, eye_movements), that may indicate other pathologies present in these datasets that our method is not designed to address. Finally, as the reviewer suggests, more sophisticated methods incorporating frequency reduction as an inductive bias, such as those accounting for interactions between features, could further improve performance in some cases.
>
> **Comment 4: “The empirical experiments are not sufficiently described in the main text; at the very least, it would be useful for the authors to clearly list which datasets are used in the experiments.”**
>
> We will add this discussion to the main text if accepted.
>
> **Minor Comments. “The main takeaways from Figure 3 are not entirely clear in the main text. IIUC the takeaway is that the CI bars are narrower in the center and right columns? For each "conclusion" the authors make about Figure 3, it would be helpful to explain how/where this is demonstrated in the figure. Also, why is there a feature with such large variance and instability in the rightmost column?"**
>
> We agree that this can be clarified further in the main text. If accepted, we will provide additional details on interpreting the figure. Different columns in the figure correspond to different weights (scale, rank, combine) for a select dataset ( electricity), and the intended message is the same for all three columns: across different random seeds (i.e., initializations), the weights converge to the same directions (the reviewer’s confidence interval interpretation is correct). This implies that the network learns similar representations across different random initializations, and does not overfit to the rest of the parameters. Connected to this, we observe that these representations can be reused across models: representations learned via MLPs can be used to improve TabNet (L343).

---

> > ### Comment · Reviewer_24qS · 2023-08-19
> >
> > Thank you for the considered response. These mostly address my concerns.
> >
> > However, I will note that the authors' claim "most datasets for which the proposed method does not achieve the highest central value AUC either already have very high baseline performance metrics for all methods (kdd-small, pol), implying that spectral bias is not as significant an issue for these datasets" *assumes* that the proposed method does what is claimed here; these experiments don't *prove* this. These empirical results do not prove, in any causal sense, that the proposed method performs best only due to improving the spectral bias and not due to any other dataset factors; I don't consider this a particularly persuasive response and still feel that more controlled (possibly synthetic data) experiments are needed to support this assumption.

---

> > > ### Author Response · Authors · 2023-08-19
> > >
> > > We thank the Reviewer for the positive score and support towards further strengthening our work, and would like to use this opportunity receive additional feedback in order to improve the camera-ready version, if accepted.
> > >
> > > We understand the Reviewer’s concern of not being able to draw causal relationships between experimental results on real-world datasets and spectral bias, as the heterogeneity of these datasets makes it challenging to rule out dataset characteristics other than target function frequency as contributing factors to performance differences. We would like to clarify that these empirical results comparing performance are primarily intended to demonstrate that our proposed method’s benefits translate to real-world datasets, complementing our analytic results that more concretely relate our proposed methods to target function frequency and spectral bias.
> > >
> > > Our reply “…most datasets for which the proposed method does not achieve the highest central value AUC either already have very high baseline…” was intended to address the Reviewer’s Comment 3, providing a possible explanation for the variation in performance improvement across datasets. We agree with the Reviewer that our comment assumes that the proposed method does what we claim (namely, mitigate the effects of spectral bias). To strengthen this assumption, and in response to the Reviewer’s suggestion in Comment 1, in our general response we provided additional experimental results on synthetic data where we varied the frequency spectrum of the target function on otherwise identical datasets. This allowed us to rule out the effects from other dataset characteristics that may be clouding the evaluation with the real-world datasets. For low-frequency target functions, all methods yield comparable results, and as target function frequency increases, the performance of all methods degrades, consistent with the expectation from spectral bias. Also, as target function frequency increases, our proposed method maintains the best performance. Assuming that the performance degradation with increasing frequency is primarily due to NN spectral bias, these results show that our proposed method is the most effective at mitigating the impact of spectral bias on NN performance, at least for these synthetic datasets.  Furthermore, the performance improvement observed with our method relative to the other baselines is significantly higher in these synthetic data experiments than that observed with the real-world datasets, consistent with the Reviewer’s expectation that differences other than target function frequency between real-world datasets can obscure the performance gains provided by our proposed method.
> > >
> > > In conclusion, we believe that the experimental results we present show that (1) our proposed method is the best among the other baselines at mitigating the impact of spectral bias and (2) the benefits of our frequency-informed methods (i.e., selective rank and the trainable layer) translate to real-world datasets. We acknowledge that though these experiments do not _prove_ that the proposed methods are improving performance solely due to frequency-reduction, they do provide compelling evidence in support of this hypothesis. Such evidence initiates a new direction to be explored in the tabular deep learning domain. We believe translating this empirical evidence into proof is a necessary follow-up study, and we are open to suggestions for other controlled experiments that can be done to further corroborate our claims.

---

### Official Review · Reviewer_uyTw · 2023-07-08

**Soundness:** 3 good
**Presentation:** 3 good
**Contribution:** 3 good
**Rating:** 5
**Confidence:** 3

**Summary:**

This paper proposes an inductive bias for tabular deep learning to bridge the performance gap between deep learning and tree-based methods on tabular data by reducing the frequency of irregular target functions through scaling and ranking transformations. Deep learning methods underperform tree-based methods on tabular data due to the interaction between irregular target functions and the tendency of neural networks to learn smooth functions. Spectral analysis tools can be used to identify the irregularity of functions described by tabular data sets and the potential for smoothing through scaling and ranking transformations. The proposed inductive bias of frequency reduction through scaling and ranking can significantly improve the performance of neural networks on tabular data without introducing additional complexity.

**Strengths:**

- The paper provides a clear and concise explanation of the performance gap between deep learning and tree-based methods on tabular data.
- The proposed inductive bias is simple and easy to implement, without requiring additional hyperparameter tuning or complex model architectures.
- The paper provides empirical evidence of the effectiveness of the proposed method on various tabular datasets and neural network architectures.

**Weaknesses:**

- The paper does not compare the proposed method with other existing methods for improving the performance of neural networks on tabular data.
- The paper does not provide a detailed analysis of the impact of scaling and ranking transformations on the interpretability of the learned models.
- The paper does not explore the potential limitations of the proposed method on highly irregular target functions or noisy data sets.

**Questions:**

 - How does the proposed method compare with other existing methods for improving the performance of neural networks on tabular data, such as feature engineering or model ensembling?
 - Can the proposed method be extended to handle highly irregular target functions or noisy data sets?
 - How does the proposed method affect the interpretability of the learned models, and how can this be addressed?

**Limitations:**

- Investigate the impact of the proposed method on the interpretability of the learned models and explore ways to improve interpretability without sacrificing performance.
- Investigate the potential limitations of the proposed method for highly irregular target functions or noisy datasets and develop alternative methods to overcome these challenges.
- Investigate the potential of combining the proposed method with other existing methods for improving the performance of neural networks on tabular data, such as feature engineering or model ensembling.

---

> ### Author Rebuttal · Authors · 2023-08-10
>
> We thank the reviewer for the insightful comments. Please see our responses below.
>
> **Q1: “How does the proposed method compare with other existing methods for improving the performance of neural networks on tabular data, such as feature engineering or model ensembling?”**
>
> We believe that the domain of tabular deep learning is in need of theoretical advancements, on top of the existing strong empirical studies (see [Related Work/Discussion/L94]). Consequently, in this work, rather than benchmarking our proposed method to compare against other existing approaches, we focus on showing that (1) NN spectral bias is an important factor contributing to the lack of performance on tabular data, and (2) learnings from this analysis can be applied to NNs straightforwardly (e.g., using selective rank (Eq. 11), or the proposed learnable convex combination (Eq. 13)).
>
> Additionally, we would like to emphasize that our proposed method is developed to introduce minimal complexity (i.e., less than a single feedforward layer) and no additional hyper-parameters, therefore, it is expected to work synergistically with other existing approaches (see Experiments/Frequency Reduction with other Network Architectures). As a follow-up study, we plan to extend our work to (1) investigate the behavior of other approaches designed towards improving neural networks on tabular data through the lens of spectral analysis, (2) evaluate the effectiveness of the proposed inductive bias on these approaches and (3) provide more extensive benchmarking.
>
> **Q2: “Can the proposed method be extended to handle highly irregular target functions or noisy data sets?”**
>
> Our method is designed to improve the NN’s ability to learn highly irregular target functions, where the irregularity is expected to convey relevant information to the corresponding task (i.e., irregularity is not due to noise). For this case, we provide additional experimental results in the PDF submitted for rebuttal, where we evaluate the performance of our method and baselines on 16 synthetic datasets with increasing target function frequency (i.e., irregularity). From the figure, we observe that our method consistently performs the best among the methods considered. Nevertheless, its performance does degrade as the frequency of the target function increases, since even with reduced relative target function frequency, the NN still is subject to spectral bias.
>
> If the highly irregular behavior of the target function is caused by noise, we expect the spectral bias of neural networks to act as a regularizer and help with generalization [1]. Although we have not explicitly studied the behavior of our method on noisy datasets, since it does not distinguish between high-frequency noise and high-frequency information, we do not expect the learned high-frequency patterns to improve generalization further. To extend our methods to noisy datasets, additional mechanisms to differentiate informative high-frequency components from noise will likely be required.
>
> [1] Fridovich-Keil, S., Gontijo Lopes, R. and Roelofs, R., 2022. Spectral bias in practice: The role of function frequency in generalization. Advances in Neural Information Processing Systems, 35, pp.7368-7382.
>
> **Q3: “How does the proposed method affect the interpretability of the learned models, and how can this be addressed?”**
>
> In this work, we focus on identifying the fundamental reasons behind the lack of performance of neural networks on tabular data. Although we believe investigating the interpretability of neural networks is important (e.g., to argue for replacing tree-based ensembles with neural networks, as they are widely accepted as more performant and interpretable alternatives), it is not in the scope of this study, however, it can be a promising future direction given the observations we make in Appendix D1 (L731). Specifically, since the proposed method independently learns a set of weights for each feature, it may be possible to interpret these weights to understand feature importances (e.g., scaling weights that converge to zero for some features may point out irrelevant/redundant features) and types of information they contribute to the decision (e.g., features with high ranking weights may indicate the feature can be compressed into quantile/percentiles instead of raw values).

---

> > ### Comment · Reviewer_uyTw · 2023-08-19
> >
> > I would like to thank the authors for their detailed response.
> > My concerns have been addressed and I will keep my score.

---

### Author Rebuttal · Authors · 2023-08-10

Dear reviewers, we thank you for all of your constructive feedback. In this general response, we would like to address some questions/concerns that arose from multiple reviewers. We address specific questions further on their corresponding threads,

**Focus of our work and its impact on the experiment design**

First, we feel it may be worth re-emphasizing the focus of our work and how it informs the metrics we choose to evaluate. The domain of tabular deep learning has been progressing rapidly, where we see extremely creative and effective ways of improving NN performance. On the other hand, there is little work that focuses on identifying the fundamental reasons behind the typical gap between tree-based and NN-based techniques on tabular data. We see our work as one of the early steps towards building a lens that can help analyzing this performance gap. We believe it opens up a new direction for researchers to study and understand the behavior of NNs on tabular data, and eventually entirely address the NN performance gap (i.e., compared to tree-based ensembles). We believe our central contribution is showing, through both theoretical and empirical analysis, that (1) NN spectral bias is an important factor contributing to the performance gap between trees and NNs, and (2) learnings from this analysis can be applied to NNs straightforwardly (using e.g., selective rank, or the proposed learnable convex combination (refer to sections)). Since we ultimately design a technique that utilizes existing transformations such as rank and scale, the expectation from these learnings is not necessarily to see unprecedented performance gains; rather, it is to see that approaches that make use of existing practices for dealing with tabular datasets (e.g., scaling, ranking) but with spectral bias in mind consistently perform better and converge faster across different datasets. In effect, the proposed approach we motivate through our analysis can be seen as alleviating the need for exhaustively searching for the correct transformation to apply to each feature in each different dataset to reduce the impact of spectral bias. For this reason, although in some instances the magnitude of performance gained on individual metrics with our proposed method appears numerically small, the overall performance improvement and convergence speed up is significant and consistent across datasets, which can be clearly seen through the normalized metrics we consider in the main text. This strongly suggests that extensions of our methods beyond the simple frequency-altering transformations we consider could provide even numerically larger gains. Note that our approach can also be used in any other NN-based method, and the benefits may vary depending on how much various architectures are impacted by spectral bias (c.f., our TabNet experiment).

**Choice of Datasets**

Second, we would like to clarify our choice of datasets for our experiments. We chose all datasets from Grinsztajn et al (NeurIPS’22), which considered several criteria for selection in order to benchmark tree-based vs NN-based model performance (Grinsztajn et al / Appendix 3). . Notably, we limited our study to numerical feature-heavy classification datasets (14) among the regression and classification datasets (45) provided in Grinsztajn et al. They are all real-world datasets that span fundamentally different problems and data characteristics, from classification for particle physics experiments (e.g., MiniBooNE) to predicting credit card defaults in Taiwan (e.g., credit card clients), containing combinations of both raw data and engineered features. Furthermore, in contrast to Grinsztajn et al, we do not remove complexities such as missing features, low/high-cardinality categorical features, and class imbalance, or discard any samples from the datasets, in order to retain the diverse properties present in these datasets

**Additional Synthetic Data Experiments**

Due to the diverse nature of the datasets considered in our experiments, there were some concerns voiced about being able to reliably draw conclusions about the effect of our proposed inductive bias from our experiments. Also, the different characteristics naturally lead to variation in the performance gain observed from our methods. To address these concerns, in this reply we also provide additional results from the synthetic data experiment in our original submission that support several of our observations made on the real-world datasets. In particular, in the attached pdf we show results obtained by varying the overall frequency of the target function in Eq. 24, keeping all other parameters fixed. For each choice of scale factor, we trained the same two-hidden-layer MLP on the raw (unit-scaled) data, and with rank, scale (standardization), and our proposed convex combination layer for 200 epochs (25 epochs were used to tune the learning rate). The results in the attached figure clearly show (1) low frequency target functions are easier for NNs to learn, (2) NN performance degrades with increasing frequency, a consequence of spectral bias, (3) our proposed method is the most robust against increased target function frequency, and (4) the difference in performance metrics (e.g., AUC) can be significantly larger numerically in datasets designed to exhibit a particular form of high-frequency behavior. Taken together, these results further substantiate our claims that frequency reduction as an inductive bias can significantly improve performance on datasets with high-frequency target functions.

---

### Decision · Program_Chairs · 2023-09-21

**Decision:**

Accept (poster)

**Comment:**

This paper introduces a novel inductive bias called "frequency reduction", reducing the frequency of irregular target functions, for improving deep learning on tabular data, to bridge the performance gap between deep learning and tree-based methods.

Four reviewers assessed this paper, and all four unanimously recommended acceptance. In particular, they all agreed on the excellence of this paper in providing a clear explanation of the performance gap between deep learning and tree-based models in tabular data and proposing an easily implementable inductive bias to bridge this gap. However, there were concerns from reviewers regarding the validation in the experiment section such as verification of simulation data and validation on higher-dimensional real datasets with diverse features and so on. I believe that including results from a variety of datasets in the final camera-ready version, including additional experiments in the rebuttal, would make this paper even stronger.